# Deconvolving mutational patterns of poliovirus outbreaks reveals its intrinsic fitness landscape

Ahmed A. Quadeer [1], John P. Barton[2], Arup K. Chakraborty[3,4,5,6,7]* & Matthew R. McKay [1,8]*

Vaccination has essentially eradicated poliovirus. Yet, its mutation rate is higher than that of viruses like HIV, for which no effective vaccine exists. To investigate this, we infer a fitness model for the poliovirus viral protein 1 (vp1), which successfully predicts in vitro fitness measurements. This is achieved by first developing a probabilistic model for the prevalence of vp1 sequences that enables us to isolate and remove data that are subject to strong vaccine-derived biases. The intrinsic fitness constraints derived for vp1, a capsid protein subject to antibody responses, are compared with those of analogous HIV proteins. We find that vp1 evolution is subject to tighter constraints, limiting its ability to evade vaccine-induced immune responses. Our analysis also indicates that circulating poliovirus strains in unimmunized populations serve as a reservoir that can seed outbreaks in spatio-temporally localized sub-optimally immunized populations.

[1] Department of Electronic and Computer Engineering, The Hong Kong University of Science and Technology, Clear Water Bay, Hong Kong, China.
[2] Department of Physics and Astronomy, University of California, Riverside, CA 92521, USA. [3] Department of Chemical Engineering, Massachusetts Institute of Technology, Cambridge, MA 02139, USA. [4] Department of Physics, Massachusetts Institute of Technology, Cambridge, MA 02139, USA. [5] Department of Chemistry, Massachusetts Institute of Technology, Cambridge, MA 02139, USA. [6] Ragon Institute of MGH, MIT and Harvard, Cambridge, MA 02139, USA. [7] Institute for Medical Engineering and Science, Massachusetts Institute of Technology, Cambridge, MA 02139, USA. [8] Department of Chemical and Biological Engineering, The Hong Kong University of Science and Technology, Clear Water Bay, Hong Kong, China. *email: arupc@mit.edu; m.mckay@ust.hk

The human immune system is incredibly effective at combating diverse pathogens. Some viruses with high mutation rates, however, have demonstrated an ability to evade immune responses through mutation while still remaining sufficiently fit to propagate infection. Prominent examples of viruses with high mutation rates include poliovirus (PV), human immunodeficiency virus (HIV), measles virus (MeV), and hepatitis C virus (HCV). PV[1] and MeV[2] are effectively controlled by vaccination, despite their high mutation rate. However, efforts continue in the quest for effective vaccines against HIV[3] and HCV[4]. A goal is to identify the features of these viruses that explain their susceptibility or resistance to control by vaccination. For the latter class (e.g., HIV, HCV), one also aims to develop an understanding of the specific sets of viral protein residues that are vulnerable to mutations, and can hence serve as targets for vaccine-induced immune responses.

Here, we focus on PV and contrast it with HIV. PV replicates in the gastrointestinal (GI) tract during the initial phase of infection. In rare cases (around 1%), PV reaches the central nervous system and may lead to poliomyelitis—a disease associated with paralysis[1]. There are two PV vaccines: the inactivated PV vaccine[5] (IPV), and the oral PV vaccine[6] (OPV). IPV induces a humoral immune response only and is known to induce weak GI immunity against wild-type PV. OPV, in contrast, elicits both cellular and humoral immune responses as it is based on a live attenuated virus which replicates in the GI tract, providing better GI immunity against wild-type PV. Both vaccines have been successful in the global fight against PV. Thanks to the Global Polio Eradication Initiative (GPEI) of the World Health Organization (WHO), PV has now been largely eradicated, with the exception of a few countries where it is still endemic[7].

An interesting aspect of PV is that its mutation rate is higher than that of HIV (~$10^{-4}$ vs. ~$10^{-5}$ substitutions per nucleotide per replication cycle[8]). This, in turn, raises the question: Why have vaccines against PV proven substantially more effective than those against HIV? HIV is well known for its ability to evade immune control, and for sustaining a large number of mutations while still remaining strongly pathogenic[9]. It may be that PV is more successfully controlled by vaccination in part because PV is under more stringent evolutionary constraints compared to HIV, limiting its ability to escape immunity through mutation to strains that are simultaneously sufficiently fit and not subject to immune attack. However, a systematic characterization of such constraints has yet to be established.

We aim to address this question by inferring the fitness landscape of PV, and subsequently comparing it with that of HIV. In previous work[10–15], we developed maximum entropy-based computational methods for inferring the fitness landscape of HIV proteins from sequence data. This involves first inferring a prevalence landscape which describes the probability of observing a sequence in circulation. For HIV, this prevalence landscape seems to serve as a rough proxy for its fitness. Several key factors have been identified to underlie this relatively simple relationship for HIV proteins[10–17]. For many HIV proteins, the inferred fitness landscapes were validated by demonstrating a high correlation of the inferred landscape-based predictions with in vitro fitness measurements and in vivo studies of virus evolution in patients[10–15]. However, applying the same methods to PV presents a challenge. All available PV sequences are sampled from the post-vaccination era—during which, as a consequence of the GPEI, it has been largely limited to the endemic regions or to outbreaks which have occurred sporadically—and it is not clear a priori whether one can disentangle intrinsic fitness effects from the biased vaccine-driven evolution in the inferred prevalence landscape in a simple way[18].

Here, we first present an analysis of the prevalence landscape of a PV protein. We focus on the capsid vp1 protein because it is the major target of neutralizing antibodies[19] and it has the highest number of available sequences (see the "Methods" section). We observe an emergent structure in the inferred prevalence landscape, which seems to disentangle the effects of intrinsic fitness and vaccine-derived selective pressures. We find that the landscape decouples into two main regions: (i) a region associated with spatio-temporally localized outbreaks, representative of PV evolution under vaccine-mediated immune pressure; and (ii) a region largely representing natural PV evolution in unimmunized populations, which is thus expected to be concordant with evolution under intrinsic constraints.

This structure is revealed through a decomposition of the inferred landscape into prevalence peaks[20,21]. Most such peaks are associated with mutations in the antigenic sites (targets of antibodies), suggesting that they represent subpopulations of viruses exhibiting immune escape. Inspection of the temporal and geographical distribution of vp1 sequences that lie on these peaks reveals that they represent spatio-temporally localized polio outbreaks. Distinct from these, a single peak which is populated by the largest number of sequences is shown to be representative of natural evolution of vp1 in largely unimmunized populations. Fitness predictions using a landscape inferred exclusively from the sequences from this peak yield an excellent correspondence with experimental fitness measurements, supporting the conclusion that it is a good proxy for the vp1 intrinsic fitness landscape. We also find that this particular peak serves as a reservoir that can transmit PV to cause outbreaks in sub-optimally immunized populations.

Comparing the obtained vp1 landscape with that for two HIV proteins—p24 (capsid) and gp160 (envelope)—suggests that PV vp1 is more constrained in its evolution than corresponding HIV proteins. Thus, despite the fact that PV has a higher mutation rate than HIV, the fitness costs of mutations appear to be more likely to be higher. This provides a partial rationale for why vaccines against PV have been so successful compared with HIV, for which immense efforts for an effective vaccine are still ongoing. Some other reasons include the fact that HIV infects immune cells[22] and has a low spike density that may limit antibody binding[23].

## Results

**Inferred vp1 landscape reflects statistics of observed data.** As a first step in inferring the fitness landscape of vp1, as established previously for HIV proteins[10–15], we sought to infer a probabilistic model, referred to as a prevalence landscape, that approximately describes the distribution of the sequences observed in the multiple sequence alignment (MSA). In this model, the probability of observing an arbitrary sequence $\boldsymbol{b} = \{\boldsymbol{b_1}, \boldsymbol{b_2}, \ldots, \boldsymbol{b_N}\}$ is given by

$$P(\boldsymbol{b}) = \frac{e^{-E(\boldsymbol{b})}}{Q}, \qquad (1)$$

where

$$E(\boldsymbol{b}) = -\sum_{i=1}^{N} h_i(b_i) - \sum_{i=1}^{N-1} \sum_{j=i+1}^{N} J_{ij}\left(b_i, b_j\right).$$

following the language of statistical physics, $E$ is referred to as the energy of the sequence. Here, $N$ is the length of the sequence, $b_i$ represents the amino acid present at the $i$th residue, and $Q = \sum_{\boldsymbol{b}} e^{-E(\boldsymbol{b})}$ is a normalization factor. The energy of a sequence is inversely related to its prevalence, and therefore a sequence with lower energy is predicted to be more prevalent (and vice versa). The energy depends on the model parameters

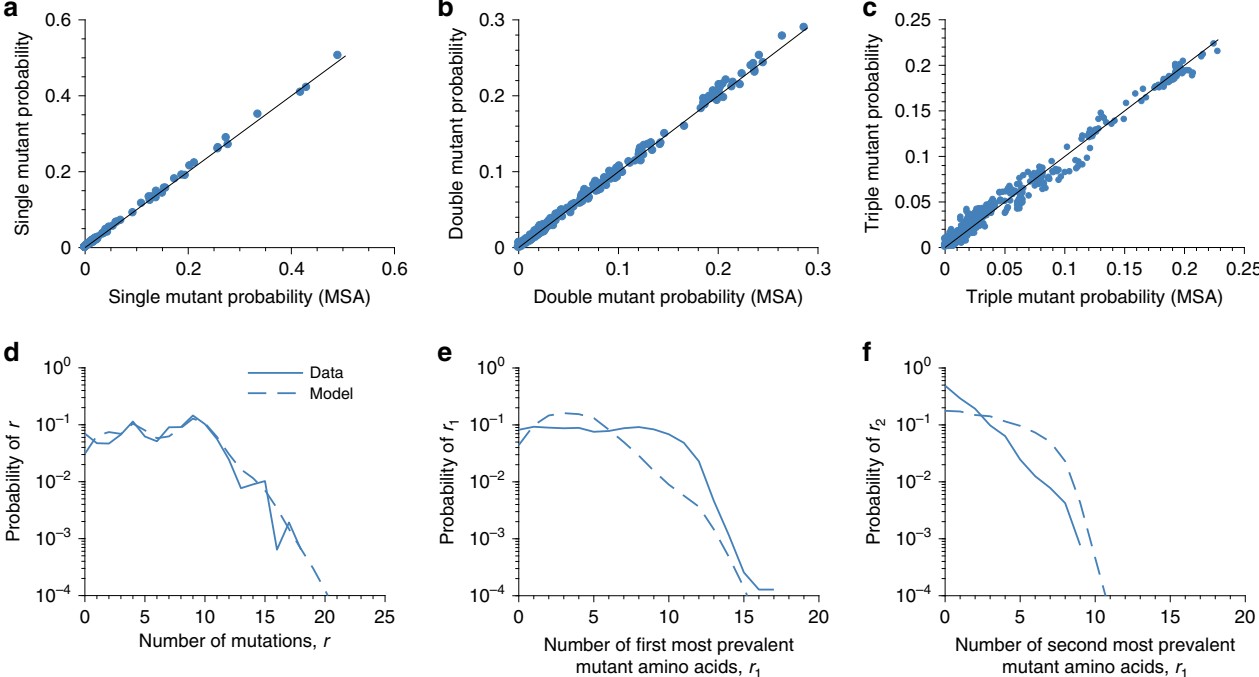

**Fig. 1 The inferred Potts model accurately captures the single and double mutant probabilities, as well as the higher order statistics of the observed sequences. a, b** Comparison of the data statistics used to train the model: **a** the single mutant probability, and **b** the double mutant probability of the observed sequences and of those obtained from the inferred Potts model. **c–f** Comparison of the data statistics predicted by the inferred Potts model: **c** the triple mutant probability, **d** the distribution of the number of mutations, **e** the distribution of the first most-prevalent mutant amino acid, and **f** the distribution of the second most-prevalent mutant amino acid in the observed sequences and the inferred Potts model. For model prediction related to connected correlations in the data, i.e., the correlations which cannot be explained by lower order mutant probabilities, see Supplementary Fig. 9.

$h_i(x)$ (called fields that represent the effects of mutations at individual residues) and $J_{ij}(x, y)$ (called couplings that represent the interaction between mutations at different residues), which are specified such that they reproduce the single and double mutant probabilities observed in the MSA (see the "Methods" section). Monte Carlo-based approaches are often used to infer these parameters, but such methods generally become computationally prohibitive for large systems[24,25]. Here, we inferred the model parameters using a recently proposed efficient approach known as adaptive cluster expansion (ACE)[24,26]. The single and double mutant probabilities of the inferred model matched well with those of the MSA, demonstrating successful model fit (Fig. 1a, b). Moreover, although not used as constraints for training the model, additional statistics computed from the inferred model—including the triple mutant probabilities, the distribution of the number of mutants, and the distribution of the number of first and second most-prevalent mutant amino acids per sequence—matched well with those of the MSA (Fig. 1c–f). These results demonstrate the accuracy by which the inferred model reflects statistical variations in the observed sequence data for vp1. We obtained prevalence landscapes with similar accuracy for the HIV proteins p24 and gp160 (Supplementary Figs. 1 and 2), as reported previously[10,11,13,14].

**Most local peaks in vp1 landscape reflect polio outbreaks**. For numerous HIV and HCV proteins, the prevalence landscapes have been shown to be predictive of intrinsic fitness[10–15,27,28]. Interestingly, such a simple correspondence between prevalence and intrinsic fitness landscape is not observed for influenza A virus (IAV), due to the evolution of IAV being driven strongly out of steady state by effective herd memory responses[29,30]. Like IAV, PV evolution is known to be under sustained and effective vaccine-mediated immune pressure, which is expected

to induce a similar significant bias in the MSA. That is, the vp1 prevalence landscape is expected to reflect a potentially complex combination of both the intrinsic and vaccine-mediated evolution of PV, and the extent to which these factors may be easily uncoupled is unclear. Here, we show that in the case of PV, intrinsic fitness constraints and vaccine-induced effects can be apparently disentangled by analyzing the structure of the inferred vp1 prevalence landscape using meta data. This is done by obtaining a low-dimensional representation of the landscape in terms of its local peaks, and performing an antigenic, geographical, and temporal analysis of the local peaks to explore their evolutionary origin. Previously, a similar analysis showed such peaks in HIV to be associated with immune escape[20]. In case of PV, this analysis allows us to suppress the bias caused by the vaccine-induced immune pressure and infer a meaningful fitness landscape model, as we discuss in the subsequent section.

We obtained a low-dimensional representation of the vp1 landscape by following a steepest ascent walk from each sequence in the MSA (see the "Methods" section). This procedure produced 25 local peaks in the vp1 landscape. We ranked these according to the number of MSA sequences that correspond to each peak, with the most-populous peak ranked number 1. This analysis revealed that the great majority of sequences (86%) lie on the top 10 peaks (Fig. 2a). An important feature of our landscape is that it is inferred by including statistical correlations between mutations at different residues. A model obtained without the effect of coupled mutations (Supplementary Note 1) results in a landscape with a single peak. We also confirmed, following a procedure similar to ref. [20], that the observed multiple local peaks in the vp1 landscape are not an artifact of finite sampling (Supplementary Note 2). Moreover, the observed peaks are found to be fairly robust to the regularization strength, or to sequence

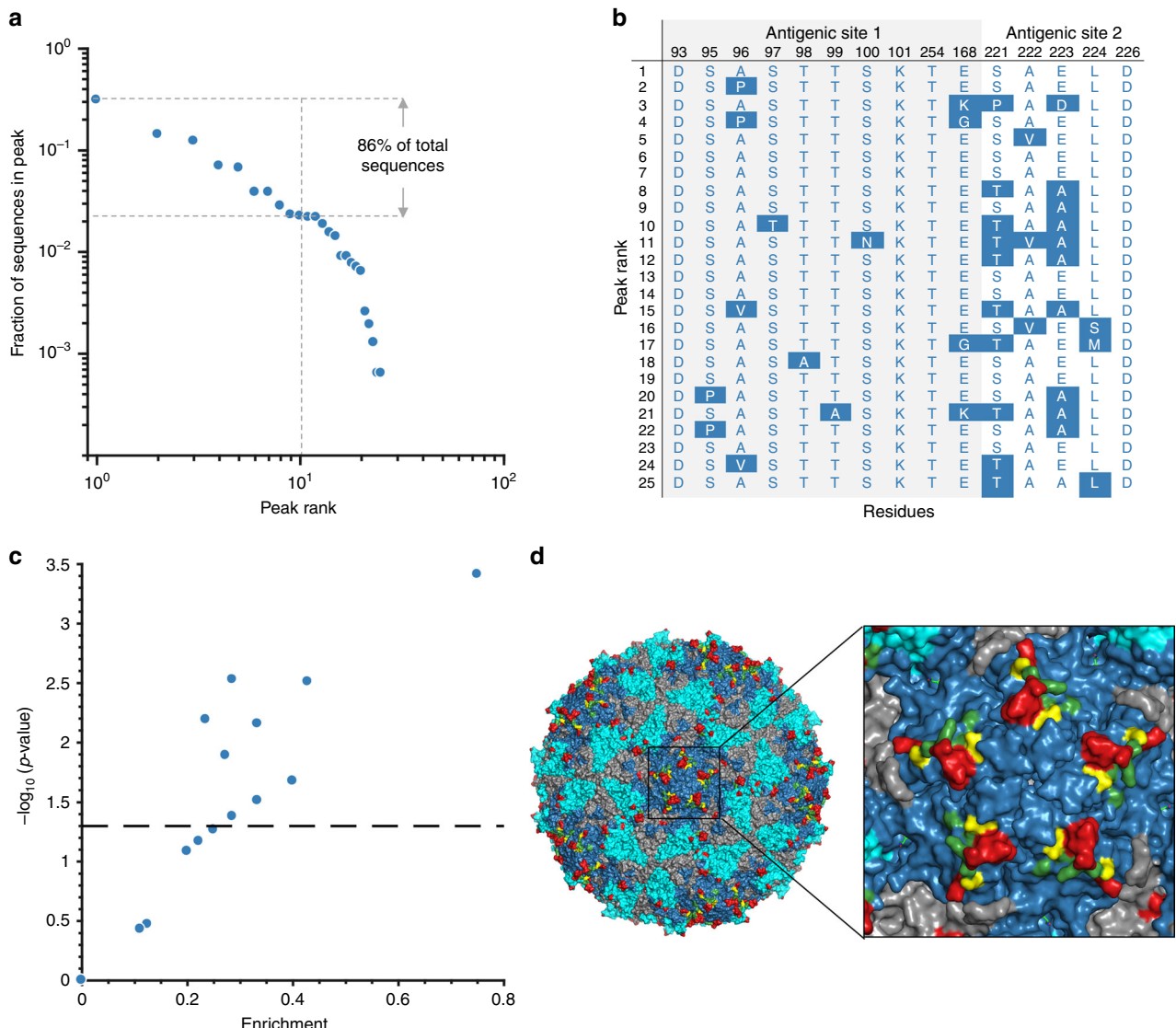

**Fig. 2 Local peaks in the prevalence landscape of vp1 protein and their antigenic analysis. a** The fraction of MSA sequences in each peak of the vp1 prevalence landscape versus rank on a log–log scale. **b** A large number of the peak sequences (16 out of 25) are enriched in mutations on the antigenic sites of vp1. These include the residues 93, 95–101, 254, 168, 221–223, 224, and 226, which are indicated in black in the top row. The combination of amino acids at antigenic sites of each peak sequence is shown in blue and the mutations compared to the consensus sequence represented in blue shaded blocks. Note that we focus here on the peak sequences only as they are a good representative of the antigenic makeup of the associated MSA sequences (Supplementary Fig. 10). **c** Out of the 16 peak sequences with mutations on antigenic sites, 10 (56%) peak sequences have statistically significant results (for *p*-value calculation, see the "Methods" section). Enrichment values are defined as the number of antigenic mutations in a peak sequence divided by the total number of mutations observed in that peak sequence. The black dotted horizontal line is the cut-off for statistically significant results ($p < 0.05$), with all the peak sequences above this line being classed as statistically significant. **d** Prediction of novel antigenic residues in vp1 protein. The biological assembly of the complete capsid protein of PV (PDB ID: 2PLV) consists of 60 copies each of vp1, vp2, vp3, and vp4 proteins. Out of these four proteins, vp1, vp2, and vp3 are on the surface of the virus as indicated by blue, cyan, and gray colors respectively. The mutations observed in the known antigenic sites[31] in the peak sequences are shown in red (residues 95–100, 168, and 221–224) while the remaining ones (residues 93, 101, 254, and 226) are shown in green. The neighboring residues of Ags1 (residues 169, 252, and 257) that were observed to have mutations in the peak sequences are shown in yellow.

reweighting used in the landscape inference procedure designed to compensate spatio-temporal biases in the PV sequence data (Supplementary Notes 3 and 4).

We investigated the local peaks by scanning the most fit sequence corresponding to each peak, referred to as a peak sequence (see the "Methods" section), for the presence of mutations at antigenic sites (groups of residues known to be targeted by neutralizing antibodies). Two out of three antigenic sites of PV are located on the vp1 protein[31]. Specifically, antigenic site 1 (Ags1) comprises three distinct segments of vp1: residues 91–102, residue 254, and residue 168; while antigenic site 2

(Ags2) consists of a segment from vp1 corresponding to residues 221–226, along with two segments of vp2. Here we defined mutation as any amino acid difference from the consensus sequence of the MSA (i.e., a sequence comprising the most frequently observed amino acid observed at each position in the MSA). Compared with the consensus sequence, we found that 72% of the peak sequences (18 out of 25) had mutations in at least one antigenic site of vp1 (Fig. 2b), suggesting the likely presence of escape mutations from natural or vaccine-induced antibodies. Of the 25 peak sequences, closer analysis revealed 16 unique sets of amino acids at antigenic sites (Fig. 2b), depicting the multiple

possible pathways employed by the virus to avoid immune pressure. Further analysis showed that for 10 of these peak sequences the antigenic-site associations were statistically significant (Fig. 2c; see the "Methods" section). We also found a few mutations in the peak sequences, at residues 169, 252, and 257, that have close physical proximity (i.e., at a distance of <8 Å) to the residues in Ags1. Mapping the known antigenic sites and these neighboring residues onto the 3D crystal structure of the PV capsid indicates that the latter are reasonably exposed on the viral surface as well (Fig. 2d). This suggests that these residues might be novel antigenic residues which contribute to escape from antibodies.

To further explore the evolutionary origin of the peaks, we conducted a geographical and temporal analysis of the vp1 sequences in each peak. Specifically, we used the country and year information, extracted from the compiled metadata of observed vp1 sequences (Supplementary Data 1), to compute the geographical and temporal distribution of the sequences in each peak. This revealed a strong association with PV outbreaks in a particular region (Fig. 3a) and time (Fig. 3b) for all but the most-populous peak 1, in which sequences from different regions at different times were present. Investigation of the literature reports associated with the sequences associated with each peak further indicates that the outbreaks associated with the majority of peaks (peak 1 being an exception) occurred due to decrease in selective pressure in largely immunized populations. We can broadly divide these peaks into three categories according to the cause reported for the representative outbreaks (Fig. 3c), as discussed below.

First, peaks 2–4, 6, 8, and 9 represented outbreaks in polio-free regions due to large gaps in immunization or incomplete immunization dosage (Fig. 3c; Supplementary Note 5). Both of these are known to cause a decrease of antibody titers in the immunized population, facilitating the occurrence of an outbreak[32,33]. Second, peak 5 consisted of a majority of sequences from the 2013 outbreak in Israel, which was seemingly caused by the exclusive use of IPV. Due to the vaccine-associated disease concern of OPV, some WHO-declared polio-free regions have switched to IPV, which is known to induce inferior (but non-zero[34]) GI immunity compared with OPV. In Israel, which adopted IPV-only dosage, circulation of wild-type PV was reported in a well-immunized (IPV-only) population (>95%) due to importation of wild-type PV from a neighboring country[35]. Third, peak 10 was associated with an outbreak caused by a wild-vaccine recombinant virus reported to circulate in China[36] during 1991–1993. Such a recombinant strain may emerge in regions where both wild-type and vaccine-derived viruses are prevalent and circulate in the immunized population due to potentially reduced immunity against it.

Note that unlike the above-mentioned peaks, peak 7 was associated with the circulation of PV in geographically localized endemic regions of Pakistan and Afghanistan[37] during 2010–2011, where a good proportion of population was unimmunized (immunization coverage was only 68–74% during this period[38,39]). Thus, it is not clear if this peak is associated with an immunized population. Nonetheless, the number of sequences in this peak was quite small, and our results were unaffected regardless of whether we associate this peak to immunized or unimmunized population (Supplementary Note 6).

In contrast to the peaks discussed above, sequences in the most-populous peak 1 were well-distributed both temporally and geographically (Fig. 3a, b). This, along with the observation of no antigenic mutations in the corresponding peak sequence (Fig. 2), suggests that peak 1 may be representative of natural PV evolution in a mainly unimmunized population. To investigate

this further we conducted a detailed analysis of the sequences in peak 1.

**Peak 1 captures evolution in largely unimmunized population.** Examination of reports in the literature associated with the sequences in peak 1 (Supplementary Table 1) revealed that it was strikingly associated with circulation of PV in largely unimmunized populations. All of the ~74% of sequences in this peak for which source information was available were obtained from either unimmunized individuals or from endemic countries, where wild PV is/was circulating in unimmunized populations. Specifically, peak 1 included sequences related to the occurrence of poliomyelitis among clinically established unimmunized children in Bulgaria[40]. It also included sequences from the endemic countries of Pakistan and Afghanistan in which wild-type PV transmission has never been interrupted[1] and sequences from the 2010 Tajikistan outbreak in which the majority of infected individuals were reported to have no detectable antibodies for PV2 and PV3 and hence were considered likely to be unvaccinated[41]. While a systematic serological study similar to refs. [41,42] was not performed for the Namibia 2006 outbreak[43] sequences present in peak 1, the majority of infected individuals were reported to be adults (born before 1990), which were most likely unvaccinated during the initial 1990–1995 OPV immunization activities in Namibia (as these specifically targeted children under 5 years of age).

Our analysis of the landscape inferred from the available vp1 sequences further suggests that the sequences in peak 1 act as a *reservoir* that transmits PV to different regions and causes outbreaks. This was established by quantifying the potential pathways for PV strains to move from one peak in the landscape to another using a zero-temperature Monte Carlo computational procedure (see the "Methods" section). This study indicated that peak 1 was distinct from other peaks as it was the most highly networked peak, with inward and outward pathways to almost all the other peaks in the landscape (Fig. 4). Interestingly, WHO has noted this potential chain of wild PV transmission from current endemic countries to the polio-free regions, and in order to stop this transmission route it has recently recommended endemic countries to ensure that all people traveling outside the country are vaccinated[44]. This result further supports the notion that peak 1 represents the evolution of PV in unimmunized population and it seems to act as a source of PV outbreaks in different parts of the world.

These results, which provide evidence that peak 1 is representative of PV in largely unimmunized populations, suggest that this part of the landscape may reflect natural PV evolution. This conclusion raises the question: Can the sequences in peak 1 provide information on the intrinsic evolutionary constraints on PV?

**Model predictions correlate well with fitness experiments.** A key goal of this study is to investigate the intrinsic fitness landscape of vp1. Based on the observations described above, we re-inferred a maximum entropy model based only on the sequences corresponding to peak 1 in order to minimize bias due to vaccine-driven evolution. We then compared fitness predictions based on this model with experimental replicative fitness measurements available from the literature[45–49].

We used the energy of a sequence computed using the inferred model (Eq. (1)), which is inversely related to prevalence/fitness, as a proxy for fitness. To test our model, we investigated whether the sequence energy values correlated negatively with corresponding experimental fitness measurements. The experimental fitness values for PV were obtained from studies[45,48–50] in which Mahoney strain variants were constructed using site-directed mutagenesis and the

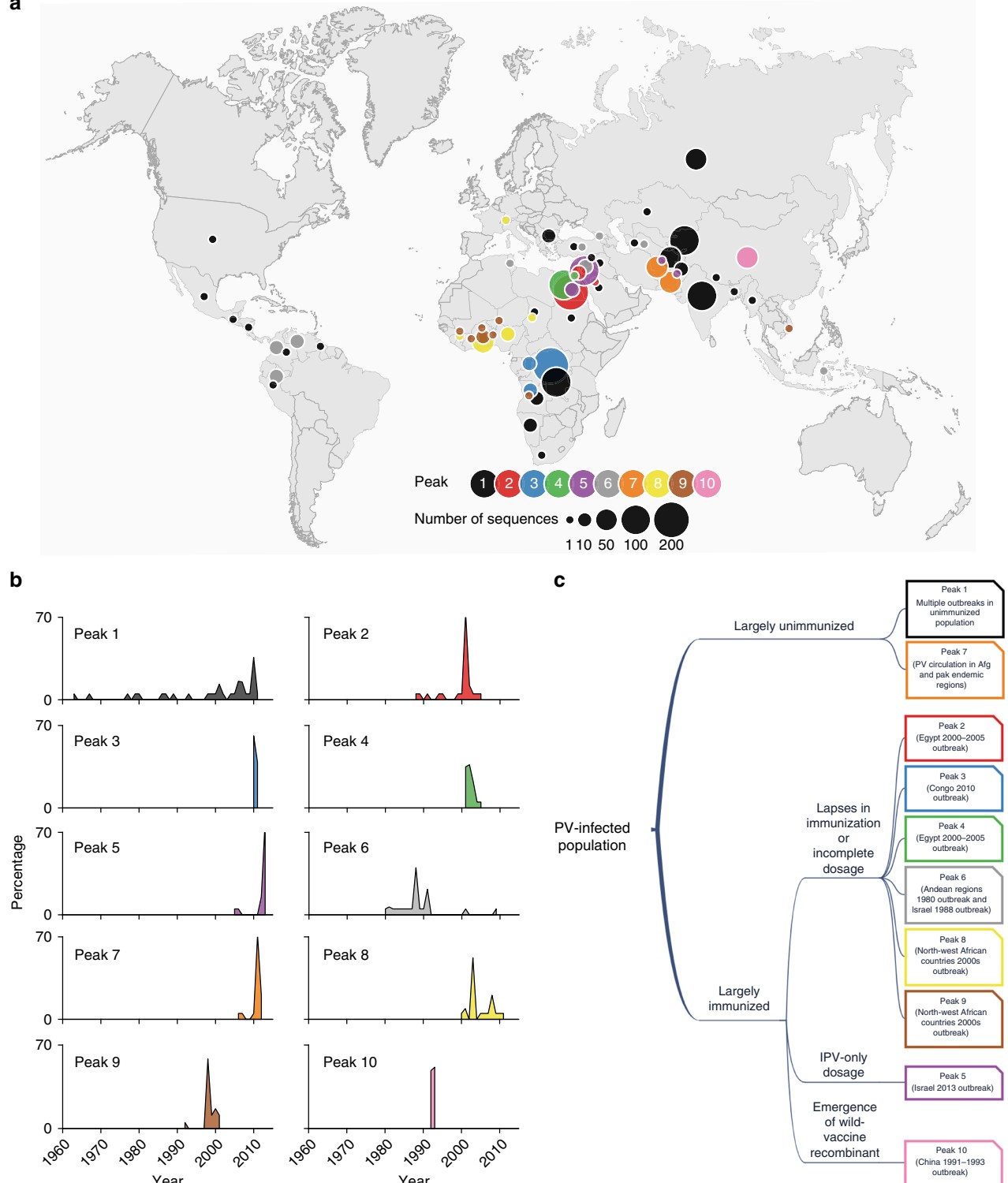

**Fig. 3 Local peak 1 in vp1 prevalence landscape is geographically and temporally well-mixed and is associated with circulation of PV in largely unimmunized populations. a** Geographical location of sequences that correspond to each peak. Only the sequences belonging to the 10 most populated peaks (~86% of the total sequences) are shown in the plot. The world map was adapted from the image available at https://freedesignfile.com/139591-simple-world-maps-vector-material-05/ which is licensed under a Creative Commons Attribution license. **b** The temporal distribution of the sequences in each of the 10 most populated peaks. Each subfigure plots the percentage of sequences in a particular peak with respect to the sampling year. The peaks are colored according to the scheme specified in **a**. **c** Association of local peaks in the landscape with largely unimmunized or immunized population based on the literature associated with the available sequence data. The outline of boxes (representing peaks) is colored according to the scheme specified in **a**.

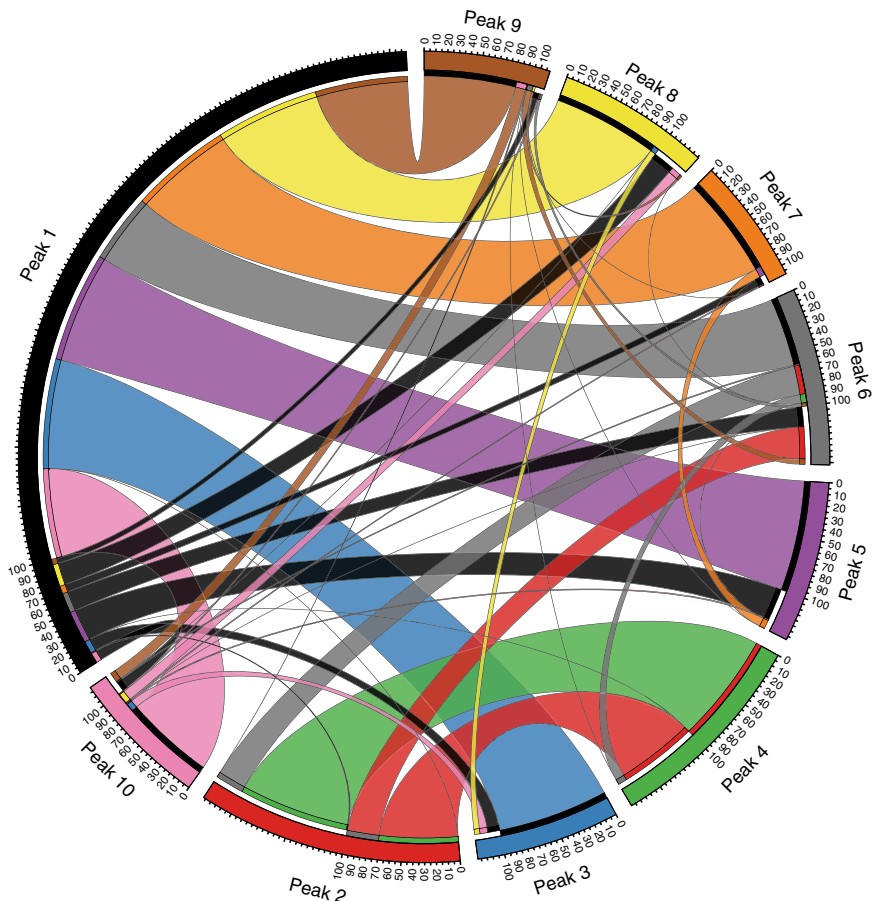

**Fig. 4 Circos plot demonstrating peak 1 to be the most densely connected peak in the inferred landscape.** A zero-temperature Monte Carlo procedure (see the "Methods" section) was used to obtain the pathways connecting different peaks in the landscape. For better visualization, only the pathways among the top 10 peaks are shown. The segments in the outer part of the circle, representing the peaks, are colored according to the scheme in Fig. 3. The ribbons, representing the pathways that connect the peaks, are colored according to the peak from which the trajectory originated. The width of each ribbon represents the percentage of the trajectories that originated from peak $k$ and reached a peak $k'$ ($k' = 1, 2, \ldots, 10, k' \neq k$) out of all the trajectories that did not reach peak $k$. This figure was generated using the web version of Circos[68].

replicative capacity of each variant was measured (see Supplementary Table 2 for the list of experimental fitness measurements and Supplementary Note 7 for the criteria used to select these measurements from each study). Comparing these fitness measurements with model energies revealed a strong negative correlation (Spearman correlation $\rho_s = -0.83$, $p = 4.5 \times 10^{-8}$; two-tailed test) (Fig. 5a). These results corroborate that our inferred model, based on peak 1 sequences only, serves as a meaningful proxy for the intrinsic vp1 fitness landscape. We note that a model inferred based on all sequences or all sequences except those in peak 1 led to a comparatively much lower correlation ($\rho_s = -0.54$, $p = 2 \times 10^{-3}$ and $\rho_s = -0.51$, $p = 3 \times 10^{-3}$, respectively) (Supplementary Fig. 4), indicating that the remaining peaks in the landscape have a strong effect of vaccine-induced immune pressure and thus, do not clearly reflect the intrinsic fitness constraints on natural PV evolution.

The strong couplings between pairs of interacting sites inferred from maximum entropy models are known to be informative about structural and functional properties of viral proteins[13,51,52] as well as other proteins[53,54]. Investigating the strongest couplings in the inferred vp1 model, corresponding to the 1% of those with the highest magnitudes, showed that 48% of these involve N-terminus vp1 residues, which form an interface with the capsid vp4 protein known to be critical for viral stability[55] (Fig. 5b). This suggests that many of the strongest couplings encode information

about constraints on the vp1–vp4 protein structure. That is, coupled mutations affect viral fitness together due to structural reasons. This result sheds light on one aspect of the biological interpretation of the intrinsic fitness landscape that we have inferred after carefully removing sampling biases due to localized outbreaks, vaccination, etc.

**Comparison with other standard methods.** The analysis above demonstrates the ability of the maximum entropy model (prevalence landscape) to distinguish meaningful clusters of sequences through its decomposition into peaks. This decomposition in particular allows us to distinguish outbreak clusters to obtain insights regarding natural virus evolution. One may ask whether such a decomposition is also revealed through phylogenetic trees or with other clustering methods which are commonly used in machine learning.

To explore this, we first constructed a phylogenetic tree (see the "Methods" section) using PASTA[56]. Coloring the sequences in the tree according to their peak identification (similar to the color scheme in Fig. 3) revealed an informative and rather consistent structure (Fig. 6a). Specifically, peak 1 sequences were largely located near the trunk of the tree, while all outbreak peaks were generally grouped in exterior branches. Consistent with Fig. 4, this result also hints at the possibility of peak 1 (representing largely unimmunized populations) being a source of all the

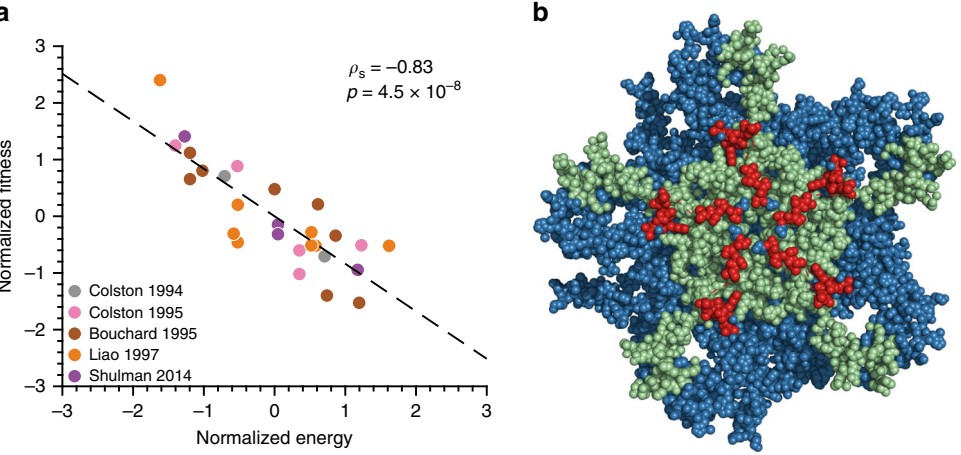

**Fig. 5 Biological validation of the inferred fitness landscape of vp1. a** In silico-predicted energy obtained from the prevalence landscape inferred from the sequences corresponding to peak 1 correlates highly with the experimental fitness measurements. Replicative fitness data was obtained from literature[45,48–50,70] and is presented in Supplementary Table 2. The black dashed line represents a linear least-squares fit to guide the eye. **b** A large proportion of the top 1 percentile model couplings are involved in forming a critical vp1–vp4 interface. The vp1 and vp4 pentamers are shown as sets of blue and green spheres, respectively, while the vp1 residues in both the top 1 percentile model couplings and in the critical vp1–vp4 interface[55] are shown as red spheres. The crystal structure of the poliovirus capsid was downloaded from the PDB database, https://www.rcsb.org/ (ID: 2PLV).

outbreaks. However, importantly, disentangling the peak 1 sequences from the outbreak peak sequences seems difficult by using just the phylogenetic tree alone (i.e., without coloring the tree according to our landscape) or even by coloring it according to spatial or temporal information available in the metadata (Supplementary Fig. 5a, b). This distinction may be due to the fact that our model takes into account the couplings between residues, while phylogenetic trees are generally constructed using a model that assumes individual residues to evolve independently. In fact, the peak structure in our inferred landscape, which is crucial to decouple the natural PV evolution from the vaccine-driven evolution (as discussed above), was not observed if we ignore the couplings between residues (only a single peak was observed in a model inferred using only residue-wise variation; see Supplementary Notes 1 and 2).

We further tested three widely used clustering methods: K-means, hierarchical, and spectral clustering[57] (see the "Methods" section). As all clustering methods are unsupervised, the number of clusters to form must be specified as an input. Thus, we tested each clustering method for two cases. For the first case, we set the number of clusters to 25, corresponding to the number of local peaks observed in the vp1 landscape in our analysis (Fig. 6b). In this case, the K-means and hierarchical clustering methods merged almost all sequences into a single cluster. In contrast, the spectral clustering method formed some clusters that were largely associated with a particular peak, suggesting that these outbreak clusters could be partially recovered using this method if the number of clusters to form is known a priori. However, the peak 1 sequences were distributed into multiple clusters, with a single cluster only capturing 37% of these sequences. Thus, the most important population for our analysis is not well captured using this approach. For the second case, the number of clusters was determined by the standard silhouette coefficient metric[58], wherein all clustering methods resulted in a maximum of three clusters, with almost all sequences falling into one of them. Thus, these results indicate that the clustering results obtained with the inferred prevalence landscape, and in particular the classification of sequences as either outbreak sequences or those representing natural evolution, is not obtained with standard clustering approaches. Beyond the PV application at hand, a

more detailed analysis of the use of maximum entropy models for clustering is a promising topic for future exploration.

**PV evolution has higher fitness constraints compared to HIV.** After validating our model as a proxy for the fitness landscape of vp1, we next compared features of this model with those of two HIV proteins: p24 and gp160. P24 is the HIV capsid protein and thus similar to vp1, which is a primary component of the PV capsid. However, unlike in HIV, the PV capsid protein is used to enter cells. Thus gp160, the envelope protein that HIV uses to attach to and enter cells and is subject to antibody responses, also serves as a meaningful comparison.

We first computed the fitness autocorrelation for each landscape, which is a statistical measure reflecting how quickly the predicted fitness varies, on average, as one walks randomly along the landscape[59] (see the "Methods" section). The decay in the fitness autocorrelation of vp1 was observed to be faster than that of HIV proteins (Fig. 7a), indicating that the fitness landscape of the surface protein of PV is relatively more constrained, i.e., taking a certain number of random steps along the PV landscape results in relatively larger changes in fitness as compared to taking the same number of random steps along the landscape of HIV proteins. The qualitative results remained the same irrespective of the sequences used to initiate the random walks (Fig. 7a).

We further compared the average neutrality of each landscape, which quantifies the average of the maximum number of mutation steps one can take without much change in fitness while performing a random walk on the landscape[59] (see the "Methods" section). Based on this measure, the landscape of vp1 appeared to be less neutral than that of both HIV proteins (Fig. 7b). This result suggests that there is generally a higher fitness cost upon making mutations in the vp1 fitness landscape as compared to the HIV proteins. This observed trend was independent of the number of steps taken in the random walk to compute the neutrality measure (Fig. 7b). We note that such a distinction in the evolutionary constraints faced by the PV and HIV proteins was not evident by using a standard dN/dS analysis (Supplementary Note 8). Moreover, such difference between the capsid proteins of PV and HIV was not obvious by using simple

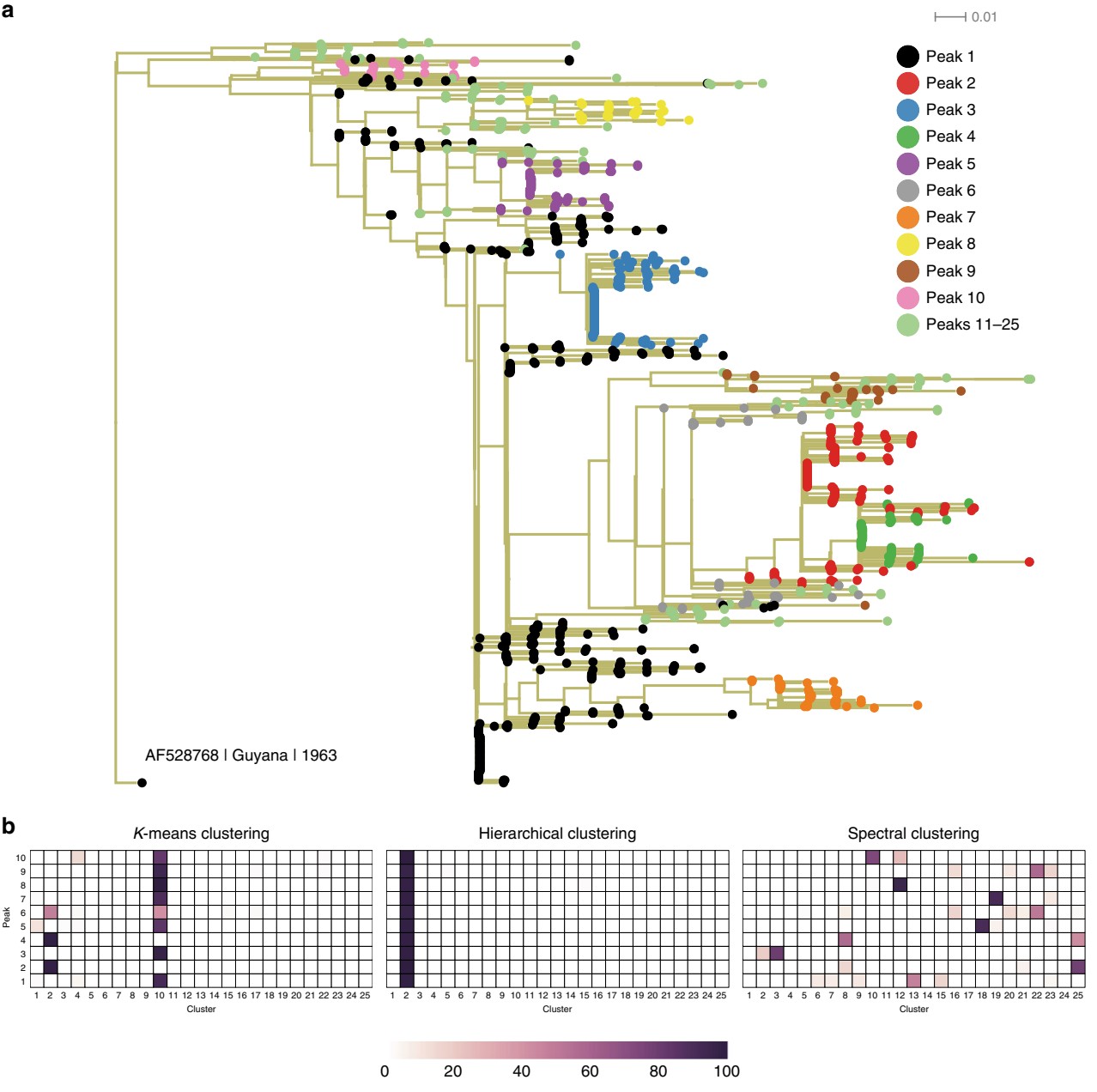

**Fig. 6 Performance (and limitations) of standard phylogenetic and machine-learning methods for distinguishing natural evolution of vp1 from vaccine-induced evolution. a** Vp1 phylogenetic tree constructed using PASTA[56] and rooted with the oldest available sequence (AF528768). The observed sequences appear at the leaves of the tree and are colored according to their peak number, while all the edges are drawn with khaki color. **b** Clusters formed using the K-means, the hierarchical, and the spectral clustering methods for the oracle case, i.e., when the number of clusters to form is set equal to the number of peaks (25) observed in the vp1 prevalence landscape. Color of a cell $(i, j)$ in the heat map represents the percentage of sequences of peak $j(j = 1, 2, ..., 10)$ that fall in a cluster $i$ formed using the specified method.

first-order analysis involving residue-wise entropies (Supplementary Fig. 6).

## Discussion

High mutation rate is an important characteristic of viruses that can enable them to evade immune responses and propagate infection. In addition, the capability of a virus to evade adaptive immunity and efficiently propagate infection is also tied to its ability to retain functionality when mutated (in other words, its ability to remain fit). Here, for the case of PV, we have shown that despite the fact that PV is highly mutable, a partial reason for the high efficacy of the PV vaccine may be due to its rigid fitness

constraints, which provide limited pathways for escaping vaccine-directed immune pressure. This was accomplished by employing computational methods to infer a fitness landscape of the PV capsid protein, vp1, and comparing it with corresponding landscapes of HIV proteins. Compared with HIV, our analysis demonstrated that PV is far less tolerant to mutations, and is under much higher intrinsic fitness constraints, making it an easier target for natural or vaccine-induced immune responses.

Inferring a meaningful fitness landscape for PV is challenging since the available sequence data reflecting circulating viral strains is strongly biased by the effects of long-standing functioning vaccines. By analyzing a low-dimensional decomposition of the inferred prevalence landscape, we showed that this bias can be

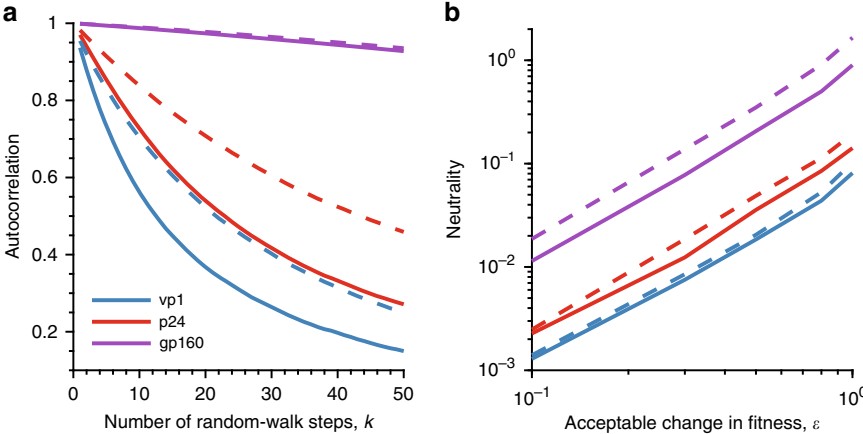

**Fig. 7 Comparison of the fitness landscapes reveals the constrained evolution of vp1 as compared to HIV proteins. a** Comparison of the autocorrelation of sequence energies of each protein landscape (see the "Methods" section). The x-axis shows the number of random-walk steps ($k$) for which the autocorrelation was computed. Results are shown for two sets of starting sequences, which were selected within a Hamming distance ($D$) of 5 (solid line) and 30 (dashed line) from the MSA (see the "Methods" section for detail). Note that the fitness landscape becomes flatter for all proteins as we allow the sequences to move further away from the MSA sequences. This is expected, since the sequences that are at a large Hamming distance from the MSA will have relatively lower fitness due to the low probability of observing those sequences (Supplementary Fig. 11). **b** Comparison of the neutrality of the landscapes of all three proteins. The neutrality measure quantifies the average maximum Hamming distance covered from an initial sequence in a neutral walk, where a random step in the walk is only accepted when the change in fitness from the sequence at the previous step is within a small value, $\epsilon$ (shown on the x-axis). For each protein, neutrality was computed for $L = 500$ (solid line) and 1000 (dashed line) steps.

significantly attenuated, producing a meaningful representative landscape of PV's evolution in largely unimmunized populations under intrinsic fitness constraints. This was validated by comparing predictions from the model with experimental fitness measurements, which showed very good agreement. This simple correspondence between the prevalence and intrinsic fitness landscape of vp1 suggests that it may also involve the key factors identified previously for this correspondence to hold in HIV proteins[16]. That is, ineffective natural immune responses against vp1 (among reported infections in unimmunized population) may be diverse, and PV may involve reversion of patient-specific immune-escape mutations to wild-type upon infection of a new patient who does not target the same region. The reported cases of infection in OPV-vaccinated patients due to reversion of the vaccine strain to wild-type[60] provides support for the involvement of the latter factor in PV infection.

In addition to identifying a large region of the inferred landscape that is reflective of intrinsic fitness of PV, investigation of the other regions of the landscape—found to reflect isolated outbreaks in largely immunized populations—offers insight into the origin and evolutionary nature of such outbreaks. While PV is generally not recognized as escaping vaccine-induced immunity through the process of antigenic drift[42] (i.e., by acquiring mutations at the antigenic sites which are the primary targets of antibodies), our analysis determined that viral strains in distinct outbreaks tend to carry antigenic mutations (Fig. 2), suggesting the potential importance of escape mutations in triggering outbreaks. These outbreak-associated strains carried other non-antigenic mutations as well which were found to be strongly coupled to the antigenic mutations (Supplementary Note 9). These non-antigenic mutations seemingly hitchhike with the antigenic mutation that potentially escapes population immunity and facilitates an outbreak in a particular region. This is reminiscent of the evolution of influenza virus[61,62], in which multiple neutral or deleterious mutations linked to an antigenic mutation, which enables escape from population immunity, have been observed to proceed to fixation collectively.

While our study has focused primarily on PV, the proposed approach of analyzing the structure of the prevalence landscape

(local peaks, etc.) may be applied more broadly for potentially isolating the intrinsic fitness landscape of viruses from strong vaccine-driven effects. MeV is one such example for which an effective vaccine has been in use for the last 50 years[2]. In a recent study, the notable effectiveness of the MeV vaccine was also suggested to be associated with the constrained evolution of MeV[63]. Moreover, it would be interesting to apply our approach to IAV, which is known to be constantly evolving under effective natural and vaccine-induced immune pressure via antigenic drift[29,64]. In contrast to spatio-temporally localized evolution of PV (Supplementary Fig. 5b), the hemagglutinin protein of IAV, which is the main target of neutralizing antibodies, is known to evolve in temporally localized clusters[64] while being geographically well-mixed (Supplementary Fig. 5c). Nevertheless, analyzing the structure of the hemagglutinin landscape would be interesting and may be informative.

## Methods

**Sequence data: acquisition and preprocessing.** All sequences available for PV were downloaded from the National Center for Biotechnology Information (NCBI) database (http://www.ncbi.nlm.nih.gov/). These sequences were then post-processed for classification into serotypes. The metadata present in the header of the downloaded sequences (in FASTA format) lacked important information, e.g., collection date, country/region, etc. Thus, an in-house code was written in MATLAB to obtain this information. In this code, the NCBI database is accessed using the accession number of the sequence and a metadata header is formed that contains the information of the collection date of the sequence, its country of origin, and the publication details (title of the report and journal name). We focused on PV serotype 1 (PV1) as it had the largest number of available sequences (Supplementary Fig. 7a). Among all serotype 1 proteins, we chose to focus on the vp1 capsid protein as it is the most exposed surface protein and is also a major target of neutralizing antibodies[19]. Analyzing the number of sequences available for each protein, vp1 also had the highest number of sequences (2261 sequences), as well as the highest ratio of number of sequences to the number of residues (Supplementary Fig. 7b–d), thus aiding statistical analyses.

The data for clade B HIV proteins p24 (8894 sequences) and gp160 (21,451 sequences) was downloaded from the Los Alamos National Laboratory HIV Sequence Database (http://www.hiv.lanl.gov). Sequences labeled as problematic or with gaps or ambiguous amino acids at ≥5% residues were removed.

All sequences for a particular protein were arranged in an $M \times N$ matrix, representing the MSA, where $M$ and $N$ denote the number of sequences and the number of residues of the protein, respectively. Each row of the MSA represents a sequence $s = \{s_1, s_2, \ldots, s_N\}$, where $s_i \in \{A, R, \ldots, V, -\}$ (the 20 amino acids and the gap). The similarity between any pair of sequences was quantified using the

similarity matrix $\boldsymbol{\Gamma}$, where the $(a, b)$th entry of $\boldsymbol{\Gamma}$ represents the similarity between two sequences $a$ and $b$ and is given by $\Gamma_{ab} = \langle \delta(s_i^a, s_i^b) \rangle_N$, where $\langle . \rangle_N$ indicates the average taken over $N$ residues, $s_i^m$ is the amino acid at residue $i$ in the $m$th sequence, and $\delta(a_1, a_2)$ is the Kronecker delta function which is equal to 1 when $a_1 = a_2$ and 0 otherwise. Principal component analysis of the similarity matrix of the vp1 protein revealed two large clusters (Supplementary Fig. 8). A detailed study of the reports associated with each sequence helped to determine that one cluster (purple) consisted primarily of wild-type sequences, while the other cluster (orange) represented the VDPV sequences. Because VDPV sequences were obtained from studies following the evolution of the vaccine strain in immunized patients, we reasoned that they were not characteristic of the typical evolution of the virus under natural constraints[65]. Thus, we focused on the wild-type 1560 sequences (Supplementary Data 1) for further analysis.

**Maximum entropy model to infer prevalence landscape.** The approach for inferring the prevalence landscape is to seek the least-biased model (i.e., the one for which the entropy is maximum[66]) that fits the single and double mutant probabilities observed in the MSA. When this approach was first applied to HIV[10], a simplified mapping was employed in which each sequence was represented by a binary string, with a '0' at a given position indicating the consensus amino acid, and a '1' indicating a mutant amino acid. With this mapping, the maximum entropy distribution took the form of an Ising model. However, while the underlying binary approximation is reasonable for conserved proteins, it results in loss of information when dealing with highly variable proteins, since it discards the identity of the mutant amino acids at each residue in the sequence. Thus, the approach has been extended to account for the mutant amino acid identities, in which case the maximum entropy distribution takes the form of a Potts model[11].

In the HIV sequence data, there is a bias due to multiple sequences per patient. Such information is not available for the PV sequence data. Examining the reports associated with these sequences (Supplementary Data 1), only a single report[67] studied viral evolution within a patient over time. Thus, we assumed each PV sequence to be sampled from a different patient. To compensate for this bias in the HIV data, a weighting was applied and the observed single and double mutant probabilities of the MSA were computed as follows:

$$p_i(x) = \frac{1}{W} \sum_{m=1}^{M} w_m \delta(s_i^m, x) \text{ and } p_{ij}(x, y) = \frac{1}{W} \sum_{m=1}^{M} w_m \delta(s_i^m, x) \delta(s_j^m, y), \quad (2)$$

where $p_i(x)$ is the frequency of observing a mutant $x$ at residue $i$, $p_{ij}(x, y)$ is the frequency of simultaneously observing a mutant $x$ at residue $i$ and a mutant $y$ at residue $j$. For the HIV proteins, the sampling bias caused by having multiple samples taken from the same patient is compensated for by the weight, $w_m = 1/z_m$, where $z_m$ is the total number of sequences in the MSA obtained from the same patient from whom sequence $m$ was acquired. The normalization factor $W$ is the number of unique patients.

We selected a limited number of mutant amino acids at each residue (instead of all 21 possibilities) such that at least 90% of the total entropy at that residue was captured. We chose this particular number to avoid fitting parameters for the rarely observed mutant amino acids. Specifically, we obtained the minimum number of mutant amino acids $a_i$ at residue $i$ such that

$$-\sum_{j=1}^{a_i} p_i(x_j) \ln p_i(x_j) \geq -0.9 \times \sum_{j=1}^{21} p_i(x_j) \ln p_i(x_j). \quad (3)$$

For the maximum entropy model, the probability of observing an arbitrary sequence $\boldsymbol{b}$ is given by[66]

$$P(\boldsymbol{b}) = \frac{e^{-E(\boldsymbol{b})}}{Q}, \quad (4)$$

where

$$E(\boldsymbol{b}) = -\sum_{i=1}^{N} h_i(b_i) - \sum_{i=1}^{N-1} \sum_{j=i+1}^{N} J_{ij}(b_i, b_j) \quad (5)$$

is referred to as the energy of the sequence $\boldsymbol{b}$ and $Q = \sum_{\boldsymbol{b}} e^{-E(\boldsymbol{b})}$ is a normalization factor. Note that $b_i$ represents the amino acid present at the $i$th residue in the sequence $\boldsymbol{b}$. The model parameters $h_i(x)$ (fields) and $J_{ij}(x, y)$ (couplings) are specified such that the single and double mutant probabilities,

$$\hat{p}_i(x) = \sum_{\boldsymbol{b}} \delta(b_i, x) P(\boldsymbol{b}) \text{ and } \hat{p}_{ij}(x, y) = \sum_{\boldsymbol{b}} \delta(b_i, x) \delta(b_j, y) P(\boldsymbol{b}), \quad (6)$$

match those of the observed data (given in Eq. (2)). The inference of these parameters is referred to as the inverse Potts problem. This inverse problem is known to have no analytical solution for systems with large dimensions, and thus one requires the use of numerical algorithms[24,25]. For this, we utilized a Potts extension[26] of the ACE method, originally proposed for the Ising model in refs. [24,25]. This code is available at https://github.com/johnbarton/ACE.

We generated samples from the predicted model using a Markov Chain Monte Carlo (MCMC) method[10]. The inferred model was first checked by making sure that the single and double mutant probabilities of the MCMC samples agree with those of the MSA. It was further validated by comparing the observed triple mutant

probabilities of the MSA with those of the MCMC samples, along with the probability of observing a sequence with $r$ mutations. The aforementioned validation tests provide a somewhat coarse-grained mutant versus non-mutant quantification of the model, however they do not capture the ability of the model to describe the specific identities of the mutants (i.e., the specific amino acids involved). Additional tests were therefore performed to characterize these features. Specifically, we compared the probability distribution of the number of first and second most-prevalent mutant amino acids observed per sequence in the MSA with the corresponding distribution in the MCMC samples.

**Characterizing the protein landscape by its local peaks.** From Eq. (1), we observe that the lower the energy of a sequence, the higher its prevalence/predicted fitness. Thus, for each inferred landscape, we investigated the local fitness maxima (or energy minima)[20,21]. Each such maximum is identified with a sequence having a fitness (energy) that is higher (lower) than all of its neighbors (those that are one mutation away in Hamming distance). We refer to such a sequence as a peak sequence. For any given starting sequence $\boldsymbol{b}$, a local maximum was found by following a steepest ascent walk. At the first step, the fitness of all neighbors of sequence $\boldsymbol{b}$ in the landscape was calculated and the neighboring sequence with the highest fitness was selected. The procedure was repeated until a sequence was reached which has higher fitness than all of its neighbors. Repeating the above procedure with each MSA sequence as the starting sequence, the number of obtained unique local maxima represented the number of local peaks in the landscape.

**Definition of statistical significance.** The statistical significance of the observed fraction of mutations on antigenic sites in each peak sequence (Fig. 2c) was quantified using a $p$-value, which is the probability of observing a result as extreme as or more extreme than the one being studied, assuming a null hypothesis were true. For example, assume that there are $j$ antigenic sites in vp1 and that a peak sequence, comprising $n$ mutations, includes mutations on $i$ of the $j$ antigenic sites. Here, the null hypothesis would be that the observed mutations on the $n$ sites in this peak sequence arose from a random selection from the $N$ sites of the protein. Assuming that the null hypothesis is true, the $p$-value is the probability that a peak sequence would include mutations on at least $i$ of the $j$ antigenic sites and is calculated as follows:

$$p = \sum_{q=i}^{\min(j,n)} \frac{\binom{j}{q} \binom{N-j}{n-q}}{\binom{N}{n}} \quad (7)$$

A low $p$-value ($p < 0.05$) would indicate that the null hypothesis is rejected and that it is unlikely that such a peak could arise from random chance.

**Quantifying the pathways between landscape peaks.** We used the zero-temperature Monte Carlo procedure to quantify the pathways between the inferred landscape peaks. In this procedure, we started multiple trajectories (600) from each MSA sequence associated to a particular peak $k$. In each step of a trajectory, a random residue in the sequence is selected for mutation. The resulting sequence is accepted if its fitness is more than the sequence in the previous step (note that this is different from the steepest ascent procedure which involves exhaustively finding the neighboring sequence with the *highest* fitness). This procedure was repeated until a sequence was reached which has higher fitness than all of its neighbors (those that are one mutation away in Hamming distance). As expected, an overwhelming majority of the trajectories (>90%) converge to the respective peak sequence, i.e., the local maximum of the peak $k$, with the remaining ones reaching a peak different from peak $k$. We recorded these latter trajectories as they represent the pathways that connect the peak $k$ to the other peaks in the inferred landscape. For visualizing these pathways, we used the web version of Circos[68] which is available at http://mkweb.bcgsc.ca/tableviewer/visualize/.

**Constructing the phylogenetic tree.** PASTA v1.6.4[56], software freely available at https://github.com/smirarab/pasta, was used to construct a maximum-likelihood phylogenetic tree (Fig. 6a) using the available vp1 data. For a fair comparison with other methods, the amino acid sequence data was provided as an input to PASTA. This software automatically selects the appropriate parameters for tree estimation based on the provided sequence data.

Dendroscope v3.5.9[69], software freely available at http://dendroscope.org/, was used for visualizing the phylogenetic tree and re-rooting it with the earliest available sequence (accession number: AF528768). The rectangular and circular phylogram layouts available in Dendroscope were used. The sequences, that form the leaves of the tree, were colored using an in-house code.

**Implementation details of standard clustering methods.** We tested whether our observed clustering of sequence data associated with largely unimmunized population is obtained with three standard clustering methods (Fig. 6b). The implementation details of these methods are as follows:

*K-means clustering*: We used the MATLAB function kmeans (Statistical and Machine Learning Toolbox) to implement this clustering method using the vp1 sequence data. Results are shown for the default distance measure of squared Euclidean distance. The qualitative results remained the same when other distance measures were used.

*Hierarchical clustering*: We used the MATLAB function linkage (Statistical and Machine Learning Toolbox) to obtain an agglomerative hierarchical cluster tree using the default distance measure of Euclidean distance. Clusters were then formed from this agglomerative tree using the MATLAB function cluster (Statistical and Machine Learning Toolbox). The qualitative results remained the same when other distance measures were used.

*Spectral clustering*: The un-normalized spectral clustering algorithm[57] was implemented in MATLAB using the fully connected graph for obtaining clusters.

All of the above methods require the number of clusters to be specified as an input. However, the optimal number of clusters is generally not known. To determine it, each method was used to construct 1–40 clusters and the MATLAB function silhouette (Statistical and Machine Learning Toolbox) was used to compute the silhouette coefficient[58] for each clustering solution. The number of clusters that maximize the silhouette coefficient was selected as optimum.

**Measures for comparing the landscape ruggedness of proteins**. We used two measures to compare the ruggedness of the inferred landscape of vp1 and various proteins of HIV[59]:

*Autocorrelation*: Autocorrelation quantifies the average change in fitness as one moves randomly along the fitness landscape. We began with $N_c = 10^6$ random sequences generated within a Hamming distance $D$ from the MSA. Using each of these sequences as a starting point, we simulated a 50-step random walk and recorded the fitness at each step. The autocorrelation of the sequence energies at the $k$th step is

$$a_k = \frac{\langle E(\boldsymbol{b}^0)E(\boldsymbol{b}^k)\rangle_{N_c} - \langle E(\boldsymbol{b}^0)\rangle_{N_c}\langle E(\boldsymbol{b}^k)\rangle_{N_c}}{\sqrt{\langle E(\boldsymbol{b}^0)^2\rangle_{N_c} - \langle E(\boldsymbol{b}^0)\rangle_{N_c}^2}\sqrt{\langle E(\boldsymbol{b}^k)^2\rangle_{N_c} - \langle E(\boldsymbol{b}^k)\rangle_{N_c}^2}} \tag{8}$$

for $k = 1, 2, \ldots, 50$, and $\boldsymbol{b}^k$ representing a sequence at the $k$th step. Recall that $\langle \cdot \rangle_{N_c}$ indicates the average taken over $N_c$ sequences. If $a_k$ decreases sharply with small increase in $k$, it indicates a rapid change in fitness along sequence trajectories on the landscape, and vice versa.

*Neutrality*: Neutrality is defined by the maximum number of mutation steps that can be taken in the landscape with a very small change ($\epsilon$) in the fitness. Using $N_c = 10^6$ random sequences as starting points, we performed a quasi-neutral random walk consisting of $L \in \{500, 1000\}$ steps on each sequence. At each step, we selected a random residue in the current sequence $\boldsymbol{b}^k$ for mutation, leading to a new sequence $\boldsymbol{b}^{k\prime}$. This mutation was accepted if the difference in fitness of the two sequences is less than some small value $\epsilon$, otherwise we selected $\boldsymbol{b}^{k+1} = \boldsymbol{b}^k$. At each step $k$, we computed the Hamming distance $d^k$ between $\boldsymbol{b}^k$ and the starting sequence $\boldsymbol{b}^0$. At the end of the random walk, we recorded the maximum Hamming distance for each trajectory. We quantified the neutrality of the landscape by the mean of the maximum Hamming distances, taken over the trajectories commencing at the $N_c$ random starting sequences, i.e., $\langle \max_k d^k \rangle_{N_c}$.

**Reporting summary**. Further information on research design is available in the Nature Research Reporting Summary linked to this article.

## Data availability
The detailed metadata compiled for vp1 sequences from the NCBI database and literature is provided in Supplementary Data 1. The experimental fitness measurements for vp1 compiled from literature are included in Supplementary Table 2.

## Code availability
All the data and scripts (mostly written in MATLAB) for reproducing the results can be found at https://github.com/ahmedaq/PV-vp1.

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

## Acknowledgements

This work was initiated upon a discussion between A.K.C. and David Baltimore. We thank Raymond Louie and Saqib Sohail for helpful discussions. This research was funded by the Hong Kong Research Grant Council General Research Fund with Project 16234716 (to M.R.M.), and a Harilela endowment (to M.R.M.). A.A.Q. was supported by the Hong Kong Ph.D. Fellowship Scheme (HKPFS) and A.K.C. was supported by the Ragon Institute of MGH, MIT, & Harvard.

## Author contributions

A.K.C. instigated the project. A.A.Q., J.P.B., A.K.C. and M.R.M. designed and performed research, analyzed the data, and wrote the paper. A.A.Q. performed the computations and generated all figures.

## Competing interests

The authors declare no competing interests.
