## [Peer Review File · Nature Communications]

Reviewers' comments:

Reviewer #1 (Remarks to the Author):

The manuscript describe the use of an entropy model to infer the fitness landscape of poliovirus type 1, and to compare it that of HIV. The authors report many different peaks in the inferred landscape that presumably correspond to different outbreaks where different constraints operated on the protein. They further report that the landscape of PV is much more constrained that that of HIV. This is clearly a very interesting topic which is relevant in the light of vaccine development efforts of HIV.

I will begin by stating that I am a strong proponent of phylogenetic modeling. I nevertheless still strongly support the use of additional approaches to study the evolution viruses. However, in this case I found the methods used to be somewhat flawed and limited in comparison to a phylogenetic approach, as described below. This leads to conclusions that are at best very easy to reach with a standard and much simpler phylogenetic analysis, and at worst conclusions that are likely flawed. Overall the paper is very hard to read and follow. I thus cannot recommend this paper for publication in a journal aimed for a wide audience. I do think that this approach has its merits and with proper correction this manuscript may be suitable for a more expert audience.

1. My strongest objection to the method presented is that it does not take into account the genetic distances among distances (branch lengths in a phylogenetic tree) and thus does not take into account sampling biases. Accordingly, a residue may be very prevalent in a large cluster (or "peak") of sequences simply since many very similar sequences were sampled for a given outbreak and they share the same recent ancestor. The authors do take into account this phenomenon when accounting for sequences sampled from the same patient, and this only accentuates how difficult this would be to account for in their model in a much more general manner. The solution (as I see it) - is to account directly for the phylogenetic tree.

2. The authors present several different peaks in their inferred fitness landscape that they claim cannot be detected using a clustering approach applied to the phylogeny. While these results are intriguing, I am very unsure what meaning they have, if at all. For example, they focus on cluster 5 that occurred in Israel. This outbreak occurred in children who did not receive OPV, which means they were effectively unimmunized in the gut. If so, as far as I see it they should have been part of cluster 1. I am thus concerned that peak 5 simply represents a cluster of sequences that are very similar since very little time (i.e., genetic distance) separates these sequences. Also other peaks are due to lapses in immunization (e.g., peaks 7,8,9) so I cannot see why they are different from peak 1. All in all I do not see evidence for the hypothesis the authors put forward whereby peaks in the landscape correspond to immunized (presumably partially immunized) versus non-immunized individuals.

3. The comparison of HIV to PV is interesting. However, HIV is a recent introduction into the human population whereas PV is not. Moreover, a simple dN/dS analysis suffices to show that gp160 in HIV undergoes extensive positive selection whereas VP1 in PV is under strong purifying selection. This is likely related to the fact that HIV is still adapting to the human population. So I do not see the merit in the approach proposed in this manuscript.

4. I do see the merit in the fact the model accounts for "coupling" that represents interactions between mutations. This is probably the strongest point of this method. If so, I would have liked to see a comparison to approaches that search for correlated evolution that DO take into account the phylogeny.

5. The paper is overall very hard to follow. This begins in the abstract. The third sentence was incomprehensible to me till I finished reading the paper. There are numbers cited in the paper that are not evident from the figures (e.g., where are the 86% cited on page 5 and referred to in Fig. 2a). Figures 1 and 2 are very hard to follow, probably figure 3 should be the first figure since it summarizes a large chunk of the main findings. I did not understand where equation 2 was derived from or what the intuition behind it is.

6. When using the zero-temperature Monte Carlo, genetic drift and neutrality are not taken into account.

7. The paper compares the fitness measurements to in vitro measurements. I have several issues here:

Some of the references cited in this section refer to the vaccine sequence. Others refer to infection in mice. The authors themselves state in the supplementary material that measurements of plaque size cannot be compared to titers. Finally, there is beautiful data (Acevedo et al . Nature 2014, doi:10.1038/nature12861) of a full-fledged genomic fitness landscape of PV1 that the author could compare with.

8. The authors cite several sites as belonging to antigenic regions, to the best of my knowledge some of these are not antigenic sites (e.g. sites 221-226, part of VP2 based on the cited paper). Please verify.

9. Figure S4 - it seems to me that the lack of correlation is simply since the number of sequences is smaller than those in peak 1. This can be tested by sampling sequences that span the same level of genetic diversity as those in non-peak 1 sequences, from the sequences in peak 1.

10. The authors removed from their analysis vaccine-derived sequences but still included the China outbreak which is a vaccine-PV recombination that affects the VP1 gene (part of it is vaccine derived).

Reviewer #2 (Remarks to the Author):

The paper by Quadeer et al. describes complex poliovirus nucleotide sequence analyses to identify differences between intrinsic fitness effects and vaccine-driven evolution. While the mathematical models used will be analysed by other reviewers, I would like to point out some issues that the authors should address before this paper can be considered for publication:

- My main concern relates to the data and analysis used to conclude that "A fitness model inferred from the latter class of sequences (PV from unimmunized populations) is in excellent agreement with in vitro fitness measurements". Data from Supplementary Table S1 were used to generate Figure 5. This is an important analysis for the paper so selecting the best data to measure in vitro fitness of poliovirus mutants is critical. Using virus titres in permissive condition does not seem adequate as very little differences are found between mutants. There were also mistakes in virus titres transcribed from the original papers as presented in Supplementary Table S1 (see image attached). In addition, values for PD50/LD50 were taken as virus titres. These correspond to 50% lethal or paralytic doses and lower values mean higher virulence. Other data should be used for measuring in vitro fitness. A literature review would be required to assess if sufficient of such data are available. Data on virus titres at high temperatures and/or paralysis induced in transgenic mice expressing the poliovirus receptor would be suitable as they correlate better with data from human isolates.

- Page 6, bottom of second paragraph: peak 7 corresponds to circulation of PV in Pakistan and Afghanistan in 2010-2011. There is no antigenic mutation in the peak sequence suggesting it may be representative of natural PV evolution but it constitutes an independent peak not aligned with Peak 1 where many other PV sequences from those two countries are classified. Why? This should be discussed.

- How good is the distinction of partially immunized populations corresponding to peaks 3, 8 and 9 and "unimmunized" populations related to the various outbreaks in Peak 1. For example, what is the difference in terms of immunisation and size of naive population between the outbreak in Congo 2011 (peak 3) and that in Namibia in 2006 (in Peak 1)?

- Page 8, bottom paragraph: phylogenetic analysis does not seem to provide the same resolution as the maximum entropy model but has phylogenetic analysis been done using just non-synonymous nucleotide mutations (or amino acid changes)?

- Page 13, second paragraph: VDPVs are not exclusively obtained from evolution of the vaccine strain in immunised patients. Circulating VDPVs transmit from person to person in populations of low immunity so they could be very useful for assessing virus evolution under natural constraints. Indeed, it would be very interesting to compare evolution of wild-type versus circulating VDPVs.

Reviewer #3 (Remarks to the Author):

The authors propose a method to infer an epistatic prevalence / fitness landscape for polioviruses protein vp1. Using this landscape, they map each observed vp1 sequence to a reference sequence defined as the local maximum in prevalence reached by steepest ascent. This mapping actually leads to a clustering of the originally more than 2000 sequences into 25 local landscape peaks, which in the paper are shown to contain highly interesting information in terms of geographical and temporal localisation of corresponding poli outbreaks. The most importing point is probably the identification of a peak, which is likely related to the natural evolution of polioviruses over about 5 decades, while other peaks are related to the constrained evolution in a vaccinated population. Complementary analysis, e.g. comparison to in-vitro fitness measurements, further underline the significance of the found subdivision.

The article is very interesting and exceptionally clear in the presentation. Results are highly interesting, and the authors have investigated very carefully that the findings are not artefacts of their landscape inference procedure from finite data, or that the same results cannot be obtained easily by standard clustering procedures using phylogeny or machine learning methods like k-means or spectral clustering. Equally, non-epistatic landscape models as broadly used in computational biology are not able to capture the clustered structure of the data, so an epistatic modeling appears to be essential for finding the presented structuring of the data.

I have a number of pretty minor remarks, which should be taken into account to further strengthen the paper:

1) The paper finds 25 prevalence peaks, and I am slightly concerned about the robustness of this finding. The authors show that the inference based on finite data itself is probably not responsible for the peaks, by using an MSA with scrambled columns, which has an identical sequence profile but leads to a single peak. However, inference from finite data has typically two opposing tendencies. Finite sampling and overfitting tend to introduce roughness and thus more local peaks into the inferred landscape, and the authors provide a good even if not rigorous argument that this is not the case. On the other hand, inference from finite data typically requires the use of regularisation, which tends to make the landscape flatter and to merge local peaks. So how can we exclude that, e.g., peaks 2 and 4 are not a finite-data induced split of a single peak, and the important peak 1 is not a merger of slightly separated peaks due to regularisation? From the point of view of inference, it may be hard to exclude these scenarios, even if the subsequent analysis underlines the significance of the clustering procedure.

As side remark, a network of inter-peak relations might be established in a non-stochastic way by looking to the minimal number of unfavourable mutations needed to overcome the prevalence valley between two peaks, or to study the stability of the peaks not only to single mutations but also to double-, triple mutations etc.

2) Concerning the enrichment of mutations in the antigenic sites, a better explication of the null model / p-value might be helpful. Is the calculation equivalent with doing a large number of random selections of residues and showing that the antigenic sites have more mutations than a random selection? Or does it include some implicit randomisation of the mutational patterns?

3) Are the statistical energies of the peak sequences very different from the natural sequences, or in between each other? Are they informative about the significance of the cluster?

4) The ρ_s with in vitro fitness measurements for the full MSA was indicated as -0.64, but it is not written if these concerns all measurements or the ones without outliers.

5) Concerning the terminology, "low-dimensional decomposition" sounds more like PCA than discrete finite clustering.

6) Concerning Fig. 1, the fitting and prediction qualities look impressive. However, to fully appreciate the power of the used epistatic model, it would be better to use connected correlations than double- or triple-mutant frequencies. To explain my point, if the single-site frequencies are biased, even in a model with independent sites there are non-trivial two- and three-site frequencies, which however can be reproduced by a profile model. Nonzero connected correlations cannot.

To conclude, I think that the presented work is very interesting and well written, and only minor revisions would be needed.

RESPONSE TO REVIEWERS' COMMENTS

Deconvolving mutational patterns of poliovirus outbreaks reveals its intrinsic fitness landscape (NCOMMS-18-08369)

We thank the reviewers for the time they devoted to provide detailed and thoughtful reviews, which have helped to improve our paper. Below, we address each of the reviewers' comments and note the changes made to the manuscript and the supplement in response (in blue).

Reviewer #1 (Remarks to the Author):

The manuscript describe the use of an entropy model to infer the fitness landscape of poliovirus type 1, and to compare it that of HIV. The authors report many different peaks in the inferred landscape that presumably correspond to different outbreaks where different constraints operated on the protein. They further report that the landscape of PV is much more constrained that that of HIV. This is clearly a very interesting topic which is relevant in the light of vaccine development efforts of HIV.

I will begin by stating that I am a strong proponent of phylogenetic modeling. I nevertheless still strongly support the use of additional approaches to study the evolution viruses. However, in this case I found the methods used to be somewhat flawed and limited in comparison to a phylogenetic approach, as described below. This leads to conclusions that are at best very easy to reach with a standard and much simpler phylogenetic analysis, and at worst conclusions that are likely flawed. Overall the paper is very hard to read and follow. I thus cannot recommend this paper for publication in a journal aimed for a wide audience. I do think that this approach has its merits and with proper correction this manuscript may be suitable for a more expert audience.

Response:

Thank you for the time and effort in reviewing our paper. In response to these comments, we have made extensive changes to the manuscript, which we believe have led to substantial improvement. Our point-by-point responses are detailed below.

- 1. My strongest objection to the method presented is that it does not take into account the genetic distances among distances (branch lengths in a phylogenetic tree) and thus does not take into account sampling biases. Accordingly, a residue may be very prevalent in a large cluster (or "peak") of sequences simply since many very similar sequences were sampled for a given outbreak and they share the same recent ancestor. The authors do take into account this phenomenon when accounting for sequences sampled from the same patient, and this only accentuates how difficult this would be to account for in their model in a much more general manner. The solution (as I see it) - is to account directly for the phylogenetic tree.*

Response:

We completely agree that accounting for sequence bias is a critical issue. This issue was indeed at the core to our study, but we also appreciate that it could be further clarified in the manuscript.

Regarding the maximum entropy method that we used for inferring the fitness landscape, while this does not explicitly use phylogenetic trees to remove sequence bias, this method has been shown to inherently suppresses phylogenetic biases by down-weighting the large eigenvalues of the involved covariance matrix, which are those most-affected by phylogeny (Qin and Colwell, 2018). Moreover, models based on maximum entropy methods have been shown to give meaningful fitness predictions for proteins of viruses like HIV (Ferguson *et al.*, 2013; Mann *et al.*, 2014; Barton, Goonetilleke, *et al.*, 2016; Butler *et al.*, 2016; Chakraborty and Barton, 2017; Louie *et al.*, 2018) and HCV (Hart and Ferguson, 2015), bacterial proteins (Figliuzzi *et al.*, 2016), and many other proteins (Hopf *et al.*, 2017).

In the case of PV, we agree that there would be an additional bias in the sampled sequences due to localized outbreaks during which a large number of similar sequences are sampled (e.g., the Congo 2010 and the Israel 2013 outbreaks). This is, in fact, the very bias that we show to exist particularly in outbreaks in largely-immunized populations (discussed in detail in response to

comment 3 below), and which negatively impacts the correspondence between the experimental fitness measurements and the landscape inferred using all available vp1 sequences (compare the results in Fig. 5 and Supplementary Fig. S4). One of the main contributions of our work is to identify this bias through antigenic, geographical, and temporal analysis of local peaks in the landscape (inferred using all available sequence data), and to subsequently *suppress* it by excluding those sequences representing peaks associated with outbreaks in immunized populations. We discuss this sequence data bias and how our method tries to account for it at multiple places in the manuscript.

The reviewer's comments do raise the interesting question of whether reweighting methods—similar to that used for reducing the bias due to multiple sequences per patient—could directly resolve bias reflected by sequence clusters caused by outbreaks etc.? While adopting a reweighting scheme that explicitly accounts for the entire phylogenetic tree is difficult in general, we ran additional tests to evaluate a natural reweighting scheme that decreases the statistical weight of PV sequences that are genetically similar and correspond to the same country and the same year. (A similar procedure, albeit based purely on genetic distance, has been applied for maximum-entropy-based models to reduce the effect of sampling bias for predicting protein contacts (Morcos *et al.*, 2011; Cocco, Monasson and Weigt, 2013).) With this sequence reweighting, the observed one- and two-point correlations of the MSA are given as

$$p_i(x) = \frac{1}{W_{\text{eff}}} \sum_{m=1}^M w_m \delta(s_i^m, x) \text{ and } p_{ij}(x, y) = \frac{1}{W_{\text{eff}}} \sum_{m=1}^M w_m \delta(s_i^m, x) \delta(s_j^m, y),$$

where $p_i(x)$ is the frequency of observing a mutant x at residue i , $p_{ij}(x, y)$ is the frequency of simultaneously observing a mutant x at residue i and a mutant y at residue j . The weight $w_m = 1/z_m$ compensates for the sampling bias, where z_m is the total number of genetically similar sequences (i.e., the sequences separated by a small Hamming distance D) in the MSA to the sequence m that are also obtained from the same country in the same year. The normalization factor $W_{\text{eff}} = \sum_{m=1}^M w_m$ is the effective number of sequences obtained after applying this weighting.

We applied this sequence reweighting for $D \leq d$ with $d \in \{0, 1, 2, 3\}$ and inferred new landscapes in each case; however, we found that it led to only minimal changes. Notably, the peak structure was very similar to the one identified without sequence reweighting (Fig. R1a), with merging of a few specific peaks associated with largely immunized populations for larger d values, while peak 1, which appears to represent natural PV evolution in largely unimmunized populations, remained virtually unaffected. Moreover, the fitness predictions based on the newly-inferred landscapes showed similar correlation with experimental fitness measurements as for the landscape inferred originally using no reweighting (Fig. R1b). These results, in general, show that attempting to account for sampling bias by simply reweighting sequences does not sufficiently compensate for the vaccine-associated bias present in the PV sequence data, and it does not address the main problem studied in our work (i.e., the inference of a meaningful PV fitness landscape, and the application of such landscape to study the intrinsic constraints of PV with respect to HIV proteins).

Importantly, other than our proposed max-entropy-based method projected on to a lower dimensional space, we are not aware of any existing method, phylogenetic tree based or otherwise, that systematically accounts for the strong sequence bias, especially given the differences in the outbreaks with respect to time, location, cause (discussed in detail in response to comment 3 below), and genetic diversity (Fig. R2), etc. Our analysis of the inferred landscape presents a strategy for doing this, since the majority of the involved local peaks were temporally and geographically localized (see Fig. 3) and identified specific strong bias due to PV outbreaks in largely immunized populations (Table R1). In contrast, the sequences in peak 1 were temporally and geographically well-mixed (Fig. 3) and were found to be significantly associated with an unimmunized population (Table R1), and thus comparatively less biased than the remaining peaks. This enabled us to predict that peak 1 is a better representative of natural PV evolution under intrinsic fitness constraints.

We have now incorporated details of this new sequence reweighting procedure and the associated results in a separate section in the supplement (Supplementary Text S4) and referred to it in the Results section in the revised manuscript.

Figure R1. Analysis of landscapes inferred after applying sequence reweighting to compensate for sampling bias. (a) Comparison of peaks identified in the landscapes with or without sequence reweighting. Results are shown for the landscapes inferred using the reweighting scheme for different values of D . The $(i,j)^{\text{th}}$ element of the matrix shows the percentage of sequences corresponding to the original peak i (identified in the landscape inferred using no reweighting) present in the j^{th} peak identified in the landscape inferred using reweighting. **(b)** Comparison of the Spearman correlation obtained between the experimental fitness measurements and the predictions based on the landscapes inferred with or without the reweighting scheme. The high correlation obtained using the landscape based on only the peak 1 sequences, which we propose to give a meaningful representation of the PV fitness landscape, is shown for reference.

Figure R2. Mean residue entropy of the sequences defining each of the top 10 peaks.

2. *I do see the merit in the fact the model accounts for "coupling" that represents interactions between mutations. This is probably the strongest point of this method. If so, I would have liked to see a comparison to approaches that search for correlated evolution that DO take into account the phylogeny¹.*

Response:

We agree that couplings are indeed important to capture mutational interactions that are critical determinants of structural and functional properties of viral proteins (Dahirel *et al.*, 2011; Hinkley *et al.*, 2011; Flynn *et al.*, 2017; Quadeer, Morales-Jimenez and McKay, 2018) as well as other proteins (Figliuzzi *et al.*, 2016; Hopf *et al.*, 2017). In fact, the structure in our inferred PV landscape (peaks), which is crucial to account for the vaccine-associated bias in the data as discussed in the previous response, is not observed if we ignore the couplings between residues (only a single peak is observed in a model inferred using only residue-wise variation; see Supplementary Text S1 and S2 for details). Thus, incorporation of couplings between residues is vital in our work for inferring a meaningful fitness landscape of PV. We now make clear this importance of incorporating couplings in our model in the Results section of the revised manuscript.

We also appreciate the request for a comparative evaluation, however, from an extensive literature survey, we did not identify any specific comparable method that accounts for phylogeny while studying correlated evolution in the sequence domain. There are indeed a few methods that search for correlated evolution and attempt to explicitly take into account phylogeny using phylogenetic trees (e.g., the evolutionary trace (ET) method (Mihalek, Reš and Lichtarge, 2004) and the multiple correspondence analysis (MCA) based S3det co-evolution method (Rausell *et al.*, 2010)); however, these methods infer co-evolution among *residues*, and they are not readily applicable to suppress the biases in the sequence data. Of course, if the reviewer is aware of directly applicable methods, we would welcome this specific input, and we would be happy to investigate such methods further.

3. *The authors present several different peaks in their inferred fitness landscape that they claim cannot be detected using a clustering approach applied to the phylogeny. While these results are intriguing, I am very unsure what meaning they have, if at all. For example, they focus on cluster 5 that occurred in Israel. This outbreak occurred in children who did not receive OPV, which means they were effectively unimmunized in the gut. If so, as far as I see it they should have been part of cluster 1. I am thus concerned that peak 5 simply represents a cluster of sequences that are very similar since very little time (i.e., genetic distance) separates these sequences. Also other peaks are due to lapses in immunization (e.g., peaks 7,8,9) so I cannot see why they are different from peak 1. All in all I do not see evidence for the hypothesis the authors put forward whereby peaks in the*

¹ As reviewer's comments 1 and 4 were related to phylogeny, we have re-ordered them here to provide a systematic response.

landscape correspond to immunized (presumably partially immunized) versus non-immunized individuals.

Response:

The identification and analysis of local peaks in the inferred landscape plays a vital role in removing the bias and inferring a representative PV fitness landscape in our work. To summarize, this analysis revealed a strong association of the majority of peaks with temporally and geographically localized outbreaks, which represented significant bias in the data (Fig. 3). In contrast, peak 1 was temporally and geographically quite well-mixed, and thus appeared comparatively less biased than other peaks (Fig. 3). To further explore the evolutionary origin of the peaks in general and to understand this uniqueness of peak 1 in particular, we performed a detailed study of the literature reports associated with sequences in each peak. This study showed that peak 1 was largely associated to PV evolution in an unimmunized population or outbreaks in regions where lack of immunization was widespread, while almost all remaining peaks were associated with outbreaks that occurred largely in immunized populations (Fig. R3). This result suggested that the peak 1 reflects natural PV evolution, which was confirmed by a strong correlation between fitness measurements and predictions of a model based *only* on the sequences corresponding to peak 1. The correlation was substantially lower if a model based on sequences lying in peaks associated with largely-immunized population was used. Moreover, we found that the identified peak structure was not readily learnt from the phylogenetic tree alone (see Fig. 6a and the corresponding Results subsection for details).

We understand the concerns of the reviewer, however, and appreciate that the association of different peaks in our inferred landscape with unimmunized or immunized population was not adequately explained in the manuscript. *In the revised version, we have updated this part of the Results section substantially to avoid potential confusion. We now make it clear that this association of local peaks to unimmunized or immunized population was done based on the immunization information of the infected population reported in the literature studies related to the sequences lying in the peaks. We have also added a new subfigure in the main text (Fig. 3c, reproduced below as Fig. R3) that graphically demonstrates this association of local peaks.* Below, we discuss some example cases related to the reviewer's comment to show how we performed this association; for details of association of all peaks, please refer to the related text in the Results section (highlighted in blue).

We first consider the peaks associated with PV outbreaks in populations that were largely immunized. While immunized populations should ideally be considered protected against wild-type PV, these outbreaks have been reported to occur due to decrease in selective pressure in the population. We can broadly divide the specific causes of such outbreaks into three categories (Fig. R3). (a) Lapses in immunization or incomplete dosage in polio-free regions, resulting in a decrease of antibody titers in the population, have been reported to lead to multiple outbreaks (Green *et al.*, 1993; El Bassioni *et al.*, 2003). This, for example, was reported to be the reason for outbreaks in Ghana and other neighboring North-Western African countries (represented by peaks 8 and 9) during the early 2000s (Odoom *et al.*, 2012b). While these countries had been declared polio free in the late 1990s, subsequent unstable political condition and shortage of vaccines hindered the immunization activities in the region. The importation of wild-type PV from the neighboring endemic country, Nigeria, enabled the virus to cause outbreaks by infecting the partially immunized population of this region. Other peaks (2, 3, 4, and 6) that represent outbreaks due to a similar cause (Fig. R3) are mentioned in the Results section of the manuscript. (b) Due to the vaccine-associated disease concern of OPV, some WHO-declared polio-free regions switched to IPV-only dosage. In Israel, which adopted such dosage, circulation of wild-type PV was reported in 2013 in a well-immunized (IPV-only) population (>95%) due to importation of wild-type PV from a neighboring country (represented by peak 5) (Shulman *et al.*, 2015). As mentioned by the reviewer, this population indeed did not receive OPV, but it still cannot be considered equivalent to an unimmunized population. This is because as compared to OPV, the gut immunity induced by IPV is less effective, but not zero, against infection. This is evidenced by the reduction in virus shedding in the IPV-only vaccinated individuals as compared to that in unimmunized individuals (see Table 49.12 in (Sutter *et al.*, 2018)). In addition, giving IPV to children with multiple previous doses of OPV has also been found to substantially boost gut immunity and reduce viral shedding (World Health Organization, http://www.who.int/SAGE_WG_Evidence22Oct2012.pdf). These studies

suggest that IPV induces selective pressure that affects viral evolution, which would be different from that induced in an unimmunized population. Thus, this inferior GI immunity of IPV allows for the infected population to serve as a reservoir for wild-type PV and enables the wild-type PV to circulate in the population, as observed in Israel in 2013. (c) Circulation of both wild-type and vaccine-derived viruses increases the chance of emergence of a wild-vaccine recombinant virus in patients infected with both viruses. Such a recombinant strain circulated in immunized population due to potentially reduced immunity against it, as reported in China during 1991-1993 (represented by peak 10) (Liu, Zheng and Zhang, 2000).

Figure R3. Association of local peaks in the landscape with unimmunized or immunized population based on the literature associated with the available sequence data. References are numbered according to the main text.

We note that the peak 7, that represented PV circulating in the endemic countries of Pakistan and Afghanistan in 2011-2013, was an exception in the temporally- and geographically-localized (and thus strongly biased) peaks. For this peak, it is not clear if it is associated to an immunized population. Nonetheless, the number of sequences in this peak was quite small, and our results were unaffected regardless of whether we associate this peak to immunized or unimmunized population (please check our detailed response to comment 2 of reviewer 2 related to peak 7).

In contrast to other local peaks, the reports associated with sequences in peak 1 strongly suggest that the infected population was largely *unimmunized*. For example, peak 1 included sequences of the occurrence of poliomyelitis among clinically-established unimmunized children in Bulgaria

(Kojouharova *et al.*, 2003). It also included sequences from the endemic countries of Pakistan and Afghanistan in which wild-type PV transmission has never been interrupted (World-Health-Organization, 2016). Peak 1 also included sequences from the 2010 Tajikistan outbreak in which the majority of infected individuals were reported to have no detectable antibodies for PV2 and PV3 and hence were considered likely to be unvaccinated (Yakovenko *et al.*, 2014). While a systematic serological study similar to (Yakovenko *et al.*, 2014) was not performed for the Namibia 2006 outbreak sequences present in peak 1 (Yusuf *et al.*, 2014), the majority of infected individuals were reported to be adults (born before 1990), which were most likely not vaccinated during the initial 1990-1995 OPV immunization activities in Namibia (as these specifically targeted children under 5 years of age).

In addition to the above, peak 1 included sequences from Congo and Angola outbreaks in 2007, for which it couldn't be confirmed if the infected population was unimmunized. These sequences were related to the outbreaks represented by peaks 8 and 9 (Odoom *et al.*, 2012b), which were suggested to have occurred due to lapses in immunization activities in these WHO-declared polio-free regions. However, principal component analysis of the similarity matrix of the sequences lying in peak 1 and other peaks (3, 8, and 9) related to Congo and Angola regions (Fig. R4) suggests that these particular sequences from Congo and Angola in peak 1 are closer to other sequences lying in peak 1 as compared to the sequences in peaks 3, 8, and 9. Based on this, it appears likely that these sequences may have been sampled from a subset of the infected population which was effectively unimmunized. Thus, while the distinction between the unimmunized population in peak 1 and the immunized population in other peaks is not perfect, occurrence of 78% (238/299) of the total unimmunized-population-associated sequences in the data in peak 1 (Table R1) provides very strong evidence that this peak is largely reflective of an unimmunized population, which is clearly not the case for the other peaks (except peak 7). Fig. R4 and Table R1 have now been included in the supplement and the related text in the manuscript has been updated.

Figure R4. Scatterplot of the first two principal components of the similarity matrix (see Methods) constructed using the sequences in peaks 1, 3, 8, and 9.

Table R1. The statistical significance of the association of each local peak with unimmunized or immunized PV-infected population.

Peak	Number of sequences			References [#]	Statistical significance of association (p-value)*	
	Unimmunized population	Immunized population	Unclear/information not available		Unimmunized population	Immunized population
1	234	83	167	3,6–11	10 ⁻⁶⁵	NS
2	0	206	16	12-14	NS	10 ⁻³⁰
3	0	191	0	15	NS	10 ⁻⁴⁴
4	0	108	1	12-14	NS	10 ⁻²³
5	5	96	3	3,16	NS	10 ⁻¹³
6	0	53	7	17,18	NS	10 ⁻⁶
7	60	0	0	3	10 ⁻⁴²	NS
8	0	36	8	19	NS	10 ⁻³
9	0	26	10	19	NS	10 ⁻¹
10	0	34	1	20	NS	10 ⁻⁷
Total	299	833	213			

*The statistical significance of the association of a peak with immunized or unimmunized population was quantified as follows. Assume that there are j sequences associated with immunized (or unimmunized) population in the available data and a peak, associated with n sequences, includes i that are a subset of these j sequences of immunized (or unimmunized) population. Here, the null hypothesis would be that the observed number of sequences associated with this peak occurred from a random selection from the M available sequences. Assuming that the null hypothesis is true, the p-value is then the probability that a peak would be associated with at least i of the j sequences associated with immunized (or unimmunized) population and is also calculated using equation 8 (see Methods). A low p-value ($p < 0.05$) would indicate that the null hypothesis is rejected and that it is unlikely that a peak associated with such immunized (or unimmunized) population could arise from random chance. NS denotes the clearly non-significant ($p > 0.1$) results.

[#]References are numbered according to the supplement.

4. *The comparison of HIV to PV is interesting. However, HIV is a recent introduction into the human population whereas PV is not. Moreover, a simple dN/dS analysis suffices to show that gp160 in HIV undergoes extensive positive selection whereas VP1 in PV is under strong purifying selection. This is likely related to the fact that HIV is still adapting to the human population. So I do not see the merit in the approach proposed in this manuscript.*

Response:

This raises a fair point of whether or not some of the main insights of our work can be revealed from a simple (standard) analysis like dN/dS? It turns out that this is not the case.

To this end, we first note that while the dN/dS analysis is suitable for application to distantly diverged sequences, it has been convincingly argued to be unsuitable for inferring positive or purifying selection using sequences sampled from a single population (Kryazhimskiy and Plotkin, 2008). In fact, for analysis within a population, it was shown that dN/dS is insensitive to the selection coefficient and $dN/dS > 1$ (or equivalently $dN-dS > 0$), the widely-accepted signature of positive selection over divergent lineages, does not hold. Thus, as the sequence data of each protein analysed in this work is from a single viral population clade/serotype, we may anticipate that dN/dS analysis may not yield a reliable estimate of overall selection.

Nevertheless, we conducted the dN/dS analysis for the considered viral proteins. Specifically, using the commonly employed MEGA7 software and selecting the well-known Li-Wu-Luo method (Li, Wu and Luo, 1985), this analysis suggested that each protein is under purifying selection (average($dN-dS$) < 0) (see Fig. R5). Similar qualitative results were obtained using another dN/dS analysis method—the modified Nei-Gojobori method.

Compared with dN/dS analysis, our work, which proposed to employ the fitness landscapes of PV and HIV for comparing the evolutionary constraints faced by these proteins appears to be more meaningful. The fitness landscapes have been demonstrated to be representative of the underlying

evolutionary constraints by contrasting against experimental fitness measurements (HIV proteins in previous reports (Ferguson *et al.*, 2013; Mann *et al.*, 2014; Barton, Goonetilleke, *et al.*, 2016; Butler *et al.*, 2016; Chakraborty and Barton, 2017; Louie *et al.*, 2018) and PV vp1 in this work), and they explicitly account for both site-wise variation and mutational interactions between sites.

We have included the above discussion in Supplementary Text S6 and referred to it in the Results section in the main text.

Figure R5. Comparison of the dN-dS values obtained for each studied protein from the codon-based test of neutrality analysis between sequences. dS and dN are the numbers of synonymous and nonsynonymous substitutions per site, respectively. The average of the dN-dS values computed over all sequence pairs in a protein using the Li-Wu-Luo method (Li, Wu and Luo, 1985) are shown. The probability of rejecting the null hypothesis of strict-neutrality ($dN = dS$) was set to 0.05 and only the significant dN-dS values were used to compute the average. All positions in a protein with less than 95% site coverage were eliminated. That is, fewer than 5% alignment gaps, missing data, and ambiguous bases were allowed at any position. Evolutionary analyses were conducted in MEGA7 (Kumar, Stecher and Tamura, 2016).

5. *The paper is overall very hard to follow. This begins in the abstract. The third sentence was incomprehensible to me till I finished reading the paper. There are numbers cited in the paper that are not evident from the figures (e.g., where are the 86% cited on page 5 and referred to in Fig. 2a). Figures 1 and 2 are very hard to follow, probably figure 3 should be the first figure since it summarizes a large chunk of the main findings. I did not understand where equation 2 was derived from or what the intuition behind it is.*

Response:

We thank the reviewer for pointing out this issue. In addition to the specific examples mentioned by the reviewer (for which we provide responses below), we have revised the complete manuscript to avoid similar potentially unclear text.

We have revised the abstract as follows:

Vaccination has essentially eradicated poliovirus. Yet, its mutation rate is higher than that of viruses like HIV, for which no effective vaccine exists. To investigate this, we inferred a maximum entropy model for the prevalence of viral protein 1 (vp1) sequences of poliovirus which enabled deconvolution of vaccine-driven effects, causing strong sequence bias, from natural evolution in largely unimmunized populations. A fitness model inferred from the latter class of sequences is in excellent agreement with experimental fitness measurements. The intrinsic fitness constraints thus derived for vp1 (a capsid protein subject to antibody responses) were compared with fitness landscapes of analogous HIV proteins. We found that vp1 evolution is subject to tighter intrinsic fitness constraints, limiting its ability to evade vaccine-induced immune responses. We also found that circulating poliovirus strains in unimmunized populations serve as a reservoir that can seed outbreaks in spatio-temporally localized sub-optimally immunized populations.

We have modified Fig. 2a now to include the percentage of sequences in the top 10 peaks.

Figures 1 and 2 are important for the statistical validation of the inferred prevalence landscape and for the antigenic analysis of the local peaks in it, respectively. We believe that both these figures are important for understanding this work; especially for the readers related to the fields of biophysics and computational biology. We have further elaborated the text related to these figures in the revised manuscript.

Equation 2 simply computes the autocorrelation between the sequence energy at the first step and that at the k th step. If the autocorrelation decreases sharply with small increase in k , it indicates a rapid change in energies along sequence trajectories on the landscape, and vice versa. Thus, this metric helps to quantify the ruggedness of the studied landscape. These details are present in the Methods section of the paper.

Modified Figure 2a. The fraction of MSA sequences in each peak of the vp1 prevalence landscape versus rank on a log-log scale. Majority of the sequences (86%) lie in the top 10 peaks.

6. *When using the zero-temperature Monte Carlo, genetic drift and neutrality are not taken into account.*

Response:

The zero-temperature Monte Carlo simulation was used simply to identify the structural properties of the inferred landscape (i.e., the existence of pathways from one peak to the other). It was not used for evolutionary simulations, and as such, evolutionary concepts such as drift and neutrality are not directly relevant to this analysis.

7. *The paper compares the fitness measurements to in vitro measurements. I have several issues here: Some of the references cited in this section refer to the vaccine sequence. Others refer to infection in mice. The authors themselves state in the supplementary material that measurements of plaque size cannot be compared to titers. Finally, there is beautiful data (Acevedo et al . Nature 2014, doi:10.1038/nature12861) of a full-fledged genomic fitness landscape of PV1 that the author could compare with.*

Response:

Regarding the first part of the reviewer's comment related to comparison with fitness measurements, we appreciate the potential confusion. A related point was raised by Reviewer 2 in his/her first comment as well. We have now updated this section in the revised manuscript accordingly.

As for the cited references being related to the vaccine sequence, we think that the reviewer is referring to (Bouchard, Lam and Racaniello, 1995) as the remaining references are not related to the vaccine sequence. In (Bouchard, Lam and Racaniello, 1995), although the Sabin vaccine strain was involved, the majority of the fitness results were reported for site-directed mutants introduced in the wild-type Mahoney strain. Thus, for validating our fitness predictions, we only used the fitness measurements reported for the Mahoney mutant strains and excluded those related to the Sabin strain. All fitness measurements used in comparing the model with fitness measurements in Fig. 5 are now clearly listed in Supplementary Table S1.

Figure R6. In silico predicted energy vs experimental fitness measurements. Both the fitness measurements and the predicted energies have been normalized using the standard procedure of subtracting the mean from each data set and dividing by its standard deviation. (a) Comparison of the energy of the prevalence landscape inferred from the sequences corresponding to peak 1 and the experimental fitness measurements. (b) Comparison of the energy of the prevalence landscape inferred from all sequences except those corresponding to peak 1 and the experimental fitness measurements.

As correctly pointed out by reviewer 2 as well in his/her first comment, the 50% paralytic dose (PD50) in mice does not seem to correlate well with that observed in human data due to the difference in the poliovirus receptor. Thus, we have removed those reports from the comparison in Figs. 5 and S4 and have now included PD50 values reported specifically for the transgenic mice expressing the poliovirus receptor. Interestingly, the updated correlation of the fitness measurements with the energy predicted from the prevalence landscape inferred from the sequences corresponding to peak 1 remained the same ($\rho_s = -0.83$) (Fig. R6a). In contrast, the correlation for the prevalence landscape inferred from all the sequences except those corresponding to peak 1 reduced from -0.64 (Supplementary Fig. S4) to **-0.51** (Fig. R6b). The high correlation of fitness measurements with predictions based on only peak 1 helps to reinforce the key finding of our work that the sequences corresponding to this peak more accurately represent the natural evolution of PV. In the revised manuscript, we have updated the text in the related Results subsection accordingly, in addition to Figs. 5 and S4.

In the second part of the comment, the reviewer suggested to compare our model predictions with the PV fitness measurements reported in (Acevedo, Brodsky and Andino, 2013), where the authors obtained a mutational fitness landscape for the PV genome using a novel next-generation-sequencing approach, called CirSeq. We appreciate the comment and attempted to make such a comparison. We dedicated significant time and effort to this, which was one reason for the lengthy amount of time in preparing this response. However, we encountered several issues in obtaining fitness results from this report, which we list below.

- a. The predicted fitness measurements for the whole PV genome were not made available, and we could not get them.

- b. We tried to generate them from the raw data which was made available. We could not do so because the raw reads need to first be processed for obtaining the base counts for each position of the reference sequence using the provided software, however, the obtained base counts were for positions greater than the length of the PV genome (7,448 bps). Thus, the obtained data was not usable. In addition, the authors did not provide the software for the method they used for inferring fitness values from the base counts.
- c. We tried to contact the corresponding author of the paper to either provide fitness values/software, or help us in resolving the above issues, but received no response.

It is noteworthy that even if the fitness measurements had been made available, it is questionable whether they would have provided meaningful validation data for our model. This is because the fitness values in (Acevedo, Brodsky and Andino, 2013) are calculated assuming no genetic interactions, while our model takes into account the effects of both point mutations and interactions between pairs of mutations.

8. *The authors cite several sites as belonging to antigenic regions, to the best of my knowledge some of these are not antigenic sites (e.g. sites 221-226, part of VP2 based on the cited paper). Please verify.*

Response:

As suggested, we have double-checked all of the vp1 antigenic sites used in our work (i.e., sites 91–102, 168, 221–226, and 254), and confirmed that these are indeed antigenic sites, as reported on page 470 of (Hogle and Filman, 1989).

9. *Figure S4 - it seems to me that the lack of correlation is simply since the number of sequences is smaller than those in peak 1. This can be tested by sampling sequences that span the same level of genetic diversity as those in non-peak 1 sequences, from the sequences in peak 1.*

Response:

We appreciate the reviewer's comment. In fact, the number of non-peak 1 sequences ($M = 1076$) is more than double the number of sequences in peak 1 ($M = 484$). Moreover, the genetic diversity (quantified in terms of residue entropy) of non-peak 1 sequences is greater than that of peak 1 sequences (Fig. R7). Thus, the relatively weak correlation of the landscape inferred using non-peak 1 sequences with the fitness measurements (Supplementary Fig. S4) seems neither due to a lack of sequences nor due to low genetic diversity. We have now updated the caption of Supplementary Fig. S4 in the revised supplement to make this point clear.

Figure R7. Comparison of the genetic diversity of the sequences in peak 1 and those in remaining peaks using residue entropy. In each box plot, the horizontal line indicates the median, the edges of the box represent the first and third quartiles, and whiskers extend to span a 1.5 inter-quartile range from the edges.

10. *The authors removed from their analysis vaccine-derived sequences but still included the China outbreak which is a vaccine-PV recombination that affects the VP1 gene (part of it is vaccine derived).*

Response:

We agree and are aware that the sequences in the China outbreak were a recombinant of a wild-type and vaccine strain. However, these sequences were kept in our analysis since, using a similarity measure (shown in Supplementary Fig. S9), the sequences representing this outbreak were closer to the wild-type sequences than the vaccine-derived ones.

Nevertheless, we also re-inferred the landscape using all available sequences except those related to this particular China outbreak and the results were largely unchanged. The sequences corresponding to each of the top 9 peaks of the resulting landscape were found to be similar to those obtained previously (Fig. R8). Specifically, the similarity of the sequences corresponding to peak 1, that we propose to be representing PV evolution among mainly unvaccinated individuals, was ~95%.

Figure R8. Comparison of the sequences corresponding to the top 10 peaks in our landscape with those corresponding to the respective peaks in the landscape inferred using all sequences except the China outbreak sequences. The $(i,j)^{\text{th}}$ element of the matrix shows the percentage of sequences corresponding to i^{th} peak in our landscape that also correspond to the j^{th} peak in the landscape inferred by excluding the China outbreak sequences.

Reviewer #2 (Remarks to the Author):

The paper by Quadeer et al. describes complex poliovirus nucleotide sequence analyses to identify differences between intrinsic fitness effects and vaccine-driven evolution. While the mathematical models used will be analysed by other reviewers, I would like to point out some issues that the authors should address before this paper can be considered for publication:

1. *My main concern relates to the data and analysis used to conclude that “A fitness model inferred from the latter class of sequences (PV from unimmunized populations) is in excellent agreement with in vitro fitness measurements”. Data from Supplementary Table S1 were used to generate Figure 5. This is an important analysis for the paper so selecting the best data to measure in vitro fitness of poliovirus mutants is critical. Using virus titers in permissive condition does not seem adequate as very little differences are found between mutants. There were also mistakes in virus titers transcribed from the original papers as presented in Supplementary Table S1 (see image attached). In addition, values for PD50/LD50 were taken as virus titers. These correspond to 50% lethal or paralytic doses and lower values mean higher virulence. Other data should be used for measuring in vitro fitness. A literature review would be required to assess if sufficient of such data are available.*

Data on virus titers at high temperatures and/or paralysis induced in transgenic mice expressing the poliovirus receptor would be suitable as they correlate better with data from human isolates.

Response

Thank you for pointing out these issues with the data employed for validation of fitness predictions based on the inferred landscape. As suggested, we have excluded studies from the validation analysis (Fig. 5) that reported the PD50 values for experiments in mice (Couderc *et al.*, 1993, 1994). We have now re-run the validation analysis using the data reported in the following experimental studies:

- a. (Bouchard, Lam and Racaniello, 1995): Virus titers of wild-type and mutant poliovirus determined by plaque assays on HeLa cell monolayers at 37 degrees Celsius as well as a slightly higher temperature of 40 degrees Celsius were reported in this study. The reviewer had mentioned that the virus titers at high temperature correlate better with data from human isolates. While the correlation of our predictions with virus titers at 37 degrees Celsius was also high, we found that the corresponding correlation for virus titers at higher temperature (40 degrees Celsius) were indeed slightly better (Fig. R9). Note that we could not find any other vp1-mutagenesis-related experimental study in the literature with virus titers reported at higher temperature. Moreover, we did not include the 50% paralytic/lethal dose (PLD50) values reported in this study as these did not seem to correlate ($\rho_s = 0.17$) with the reported titers for strains with the same mutations (Fig. R10).
- b. (Colston and Racaniello, 1994): The growth of wild-type and mutant poliovirus on cells expressing wild-type receptor at 37 degrees Celsius were reported in this work. Only the average values of titers were reported as all mutants were found to have similar titers.
- c. (Colston and Racaniello, 1995): The growth of wild-type and mutant poliovirus on cells expressing wild-type receptor at 37 degrees Celsius were reported in this work. The typographical error mentioned by the reviewer in the titer corresponding to the mutation P95T has now been corrected.
- d. (Liao and Racaniello, 1997): The growth of $\Delta 9$ (Mahoney strain with B-C loop deleted) and $\Delta 9$ mutants on cells expressing wild-type receptor at 37 degrees Celsius were reported in this study. The reviewer correctly pointed out that the background strain in this study was $\Delta 9$ (and not the Mahoney strain) and the titer corresponding to it was 8.5×10^8 PFU/ml (mentioned in Table 2 in (Liao and Racaniello, 1997)). We like to elaborate that the titer 2.5×10^7 PFU/ml mentioned by the reviewer was for the strain $\Delta 9$ -1/i, that comprises three additional mutations mentioned in Table 3 in (Liao and Racaniello, 1997), which was not the background strain on which site-directed-mutagenesis was performed.
- e. (Shulman *et al.*, 2015): 50% paralytic dose (PD50) values for experiments in transgenic mice expressing the poliovirus receptor were reported in this recent study. As correctly suggested by the reviewer, the smaller the PD50 value, the more is the virus's virulence. Thus, for

consistency with the viral titers and growth data from other reports, we used the inverse of the values reported for PD50 in (Shulman *et al.*, 2015).

The Supplementary Table S1 (referred to as Table S2 in the revised manuscript) has been revised accordingly (reproduced below for convenience). Interestingly, the correlation of the fitness measurements reported in these studies with the energy predicted from the landscape inferred from the sequences corresponding to peak 1 remains the same ($\rho_s = -0.83$) (Fig. R6a; reproduced below for convenience). However, the correlation for the landscape inferred from all the sequences except those corresponding to peak 1 decreases from $\rho_s = -0.64$ (Supplementary Fig. S4) to $\rho_s = -0.51$ (Fig. R6b). The high correlation of fitness measurements with predictions based on only peak 1 helps to reinforce the key finding of our work that the sequences corresponding to this peak represent the natural evolution of PV. We have now updated the related text in the Results section of the manuscript, Fig. 5, Supplementary Table S2, and Supplementary Fig. S4 accordingly.

Figure R9. In silico predicted energy vs the virus titers reported in (Bouchard, Lam and Racaniello, 1995). Results for virus titers determined at (a) 37 degrees Celsius and (b) 40 degrees Celsius.

Figure R10. Normalized virus titers vs normalized PLD50 values reported in (Bouchard, Lam and Racaniello, 1995) for the same viral strains.

Table S2. Experimental fitness values reported in (Colston and Racaniello, 1994, 1995; Bouchard, Lam and Racaniello, 1995; Liao and Racaniello, 1997; Shulman *et al.*, 2015).

(Colston & Racaniello 1994)	Virus ^a	Titer (PFU/ml) ^b
	Mahoney	4.7 x 10 ⁷
	G225D	~4.4 x 10 ⁴
	D226G	~4.4 x 10 ⁴
	D226N	~4.4 x 10 ⁴
	L228F	~4.4 x 10 ⁴
	A231V	~4.4 x 10 ⁴
	L234P	~4.4 x 10 ⁴
	D236G	~4.4 x 10 ⁴
	M132I	~4.4 x 10 ⁴
	A241V	~4.4 x 10 ⁴
	A241T	~4.4 x 10 ⁴
	H265R	~4.4 x 10 ⁴
(Colston & Racaniello 1995)	Virus	Titer (PFU/ml)
	Mahoney	1.6×10 ⁹
	P95S	1.8×10 ⁹
	V160I	7.8×10 ⁸
	P95S, V160I	8.3×10 ⁸
(Bouchard et al., 1995)	Virus	Titer (Plaque size) ^c
	Mahoney	0.315
	T36A	0.333
	T88A	0.125
	M90I	0.288
	P95S	0.380
	T99K	0.232
	A106T	0.348
	L134F	0.112
(Liao & Racaniello 1997)	Virus	Titer (PFU/ml) ^d
	Mahoney (Δ9)	8.5×10 ⁸
	V160I	2.3×10 ⁸
	W170R	4.3×10 ⁷
	T177S	8.6×10 ⁷
	V160I, W170R	2.6×10 ⁷
	V160I, T177S	2.8×10 ⁷
	W170R, T177S	9.3×10 ⁷
	V160I, W170R, T177S	2.6×10 ⁷
(Shulman et al., 2014)	Virus	50% paralytic dose (PD ₅₀) ^e
	Mahoney	5.9
	VP1_8062-PL1_ISR13	7.2
	VP1_8149-PL1_ISR13	6.7
	VP1_8150-PL1_ISR13	6.8

^a For each site-directed mutant in the Mahoney strain, the first and last letter represents the wild-type and mutant amino acid respectively, while the number represents the position of the residue in the vp1 protein. Mutations which are not seen in the MSA were replaced with the least observed mutant at that residue to predict the energy. Note that our model is not trained to predict the energy of a strain with mutation at a fully conserved residue in the MSA. As these mutants are not present in the MSA, we can assume their fitness to be very small; the fitness of strains with these mutants was found to be smaller than the Mahoney strain in the experiments as well. Thus, we set the energy of such strains to nh_{\min} , where n is the number of fully conserved residues and h_{\min} is the minimum value of the fields predicted in the inferred landscape (equation 1).

^b Only the average titers were reported for the strains with mutations in (Colston and Racaniello, 1994). The corresponding predicted energies were also averaged in the comparison in Fig. 5.

^c Instead of PFU/ml, the viral growth in (Bouchard, Lam and Racaniello, 1995) was reported in terms of plaque size. The values reported for 40 degrees Celsius were used. Note that this choice of measure of viral growth is

irrelevant for our analysis as we are interested in the relative growth of mutant strains with respect to the reference strain used in each report.

^d Instead of Mahoney strain, $\Delta 9$ —a B-C loop (residues 98–102) truncated Mahoney strain—was used in (Liao and Racaniello, 1997).

^e Only those strains were included in the analysis for which accession number was provided in (Shulman *et al.*, 2015). The PD₅₀ value reported for the Sabin 1 strain was excluded as our model is specific to the wild-type PV. In contrast to the viral titers and growth data from other reports, the smaller the PD50 value, the more is the virus's virulence. Thus, for consistency, we used the inverse of the reported values of PD50.

Figure R6. In silico predicted energy vs experimental fitness measurements. Both the fitness measurements and the predicted energies have been normalized using the standard procedure of subtracting the mean from each data set and dividing by its standard deviation. For (Bouchard, Lam and Racaniello, 1995), the virus titers reported at higher temperature of 40 degrees Celsius were used. **(a)** Comparison of the energy of the prevalence landscape inferred from the sequences corresponding to peak 1 and the experimental fitness measurements. **(b)** Comparison of the energy of the prevalence landscape inferred from the all sequences except those corresponding to peak 1 and the experimental fitness measurements.

2. Page 6, bottom of second paragraph: peak 7 corresponds to circulation of PV in Pakistan and Afghanistan in 2010-2011. There is no antigenic mutation in the peak sequence suggesting it may be representative of natural PV evolution but it constitutes an independent peak not aligned with Peak 1 where many other PV sequences from those two countries are classified. Why? This should be discussed.

Response:

This is a fair point. It is true that unlike other reported outbreak peaks, there was no antigenic mutation in peak 7 (Fig. 2b). However, it was geographically and temporally localized as it represents the circulation of PV in Pakistan and Afghanistan during 2010-2011 (Fig. 3). Thus, it is not clear if this peak is associated to a specific outbreak, particularly because both Pakistan and Afghanistan are endemic countries. The report associated with the majority of sequences in this peak (Shaukat *et al.*, 2014) appeared to be a random surveillance study of wild-type PV circulation near the Pakistan-Afghanistan border; suggesting that similar to peak 1, peak 7 may also represent the circulation of PV in unimmunized population. We tried to investigate this in detail using the available data as discussed below.

The phylogenetic tree (Supplementary Fig. S5a, reproduced below for convenience) shows that peak 7 sequences spurred out of peak 1 sequences also belonging to South Asia (Pakistan and Afghanistan) (Supplementary Fig. S5b, left panel) and thus, peak 7 may be an extension of peak 1. If this is true, we hypothesized that peak 7 should be closer to peak 1 as compared to other outbreak peaks. However, investigating the Hamming distances between the sequences in peak 7

and those in peak 1 did not show peak 7 to be the closest to peak 1 (Fig. R11a). It appeared to be the second-closest to peak 1, with peak 3 being the closest. We also tried to quantify the possible close proximity of peak 7 to peak 1 by running a zero-temperature MCMC simulation, which involved starting multiple trajectories from each sequence in peak 1 and determining the number of mutation steps required to reach other peaks. The smaller the number, the closer a peak would be to peak 1. The number of steps required to reach peak 7 was again found to be the second-smallest after peak 3 (Fig. R11b). Interestingly, peak 7 and peak 5 (representing Israel 2013 outbreak) get merged into a single peak if landscape is inferred by re-weighting of sequences for suppressing sampling bias in the data (Fig. R1). This suggests that the sequences in peak 5 are closely related to those in peak 7, which is consistent with the fact that the wild-type PV that circulated in a well-immunized population of Israel in 2013 was imported from South Asia (Shulman *et al.*, 2015). Thus, similar to peak 1, peak 7 may also be considered a reservoir of PV sequences in largely unimmunized populations that can be transmitted to cause outbreaks in regions with a sub-optimally immunized population (see Results for details).

Due to these ambiguities related to peak 7, we also computed the correlations with experimental fitness values by considering both peaks 1 and 7 to be representative of the unimmunized population (Fig. R11c) and the remaining peaks to be representative of outbreaks (Fig. R11d). The correlation values for both cases (-0.82 and -0.50, respectively) remained almost the same as compared to Fig. 5 (-0.83 and -0.51, respectively).

We have now included this discussion related to local peak 7 in the Supplementary Text S5.

Figure S5. Rectangular phylogram of the vp1 phylogenetic tree. Sequences are colored according to their (a) peak number (similar to Fig. 6a in the main text), (b, left panel) geographical information, and (b, right panel) temporal information.

Figure R11. Analysis of local peak 7. (a) Hamming distance of all sequences in each peak from sequences in peak 1. (b) Number of mutation steps required to reach a peak starting from sequences in peak 1. 600 zero-temperature MCMC runs were started from each sequence belonging to peak 1 and the peak reached at the end of each trajectory (total steps in each MCMC run = 5×10^5) was recorded. (c-d) In silico predicted energy vs the experimental fitness measurements. Both the fitness measurements and the predicted energies have been normalized using the standard procedure of subtracting the mean from each data set and dividing by its standard deviation. For (Bouchard, Lam and Racaniello, 1995), the virus titers reported at higher temperature of 40 degrees Celsius were used. (c) Comparison of the energy of the prevalence landscape inferred from the sequences corresponding to peak 1 & 7 and the experimental fitness measurements. (d) Comparison of the energy of the prevalence landscape inferred from the all sequences except those corresponding to peak 1 & 7 and the experimental fitness measurements.

3. *How good is the distinction of partially immunized populations corresponding to peaks 3, 8 and 9 and “unimmunized” populations related to the various outbreaks in Peak 1. For example, what is the difference in terms of immunisation and size of naïve population between the outbreak in Congo 2011 (peak 3) and that in Namibia in 2006 (in Peak 1)?*

Response:

The distinction between the level of immunization of infected populations associated with different peaks was performed based on the information reported in the associated literature studies. Note that the size of the naïve population in specific outbreaks was not explicitly reported in these studies. In the following, we provide details of the reported immunization level in the population affected by outbreaks corresponding to peak 1 and the few other peaks mentioned by the reviewer; for details of other peaks, please refer to the related text in the Results section (highlighted in blue).

We first consider the peaks that comprised sequences corresponding to outbreaks in immunized populations. Peak 3 was categorized as one such peak, as it represented the 2010 Congo outbreak in which the infected population was clearly reported to be immunized. This was confirmed by the presence of neutralizing antibody titers against all Sabin strains (Sabin-1, -2, and -3) in the reported fatal poliomyelitis cases (Drexler *et al.*, 2014). This outbreak had a rare (very high) mortality rate of 47% due to a mutation in the antigenic sites, which was able to evolve due to the compromised immunity of the population—a consequence of incomplete OPV dosage. Similarly, peaks 8 and 9 comprised sequences corresponding to the last-decade outbreaks in Ghana and other neighbouring North-western African countries that were reported to have occurred due to inadequate immunization activities during the early 2000s (Odoom *et al.*, 2012a). While these countries had been declared polio free in the late 1990s, subsequent unstable political condition and shortage of vaccines hindered the immunization activities in the region. The importation of wild-type PV from the neighbouring endemic country, Nigeria, enabled the virus to cause outbreaks by infecting the partially immunized population of this region.

In contrast, the reports associated with sequences in peak 1 were strongly suggestive of the infected population to be mainly unimmunized. For example, while a systematic serological study similar to (Drexler *et al.*, 2014) was not performed for the 2006 Namibia outbreak sequences present in peak 1 (Yusuf *et al.*, 2014), the majority of infected individuals were reported to be adults (born before 1990), which were most likely not vaccinated during the initial 1990-1995 OPV immunization activities in Namibia (as these specifically targeted children under 5 years of age). Similarly, the majority of infected individuals in the 2010 Tajikistan outbreak sequences present in peak 1, in contrast to those in the Congo 2010 outbreak (Drexler *et al.*, 2014), were reported to have no detectable antibodies for PV2 and PV3 and hence appeared unvaccinated (Yakovenko *et al.*, 2014).

We understand that these details were not discussed sufficiently in the manuscript. In the revised version, we have now updated the related part of the Results section substantially (highlighted in blue) to elaborate on the immunization level of populations associated with each peak, where possible. In addition, we have included a new subfigure in the main text (Fig. 3c; see Fig. R3) that graphically demonstrates the association of different populations with the local peaks observed in the inferred landscape.

4. *Page 8, bottom paragraph: phylogenetic analysis does not seem to provide the same resolution as the maximum entropy model but has phylogenetic analysis been done using just non-synonymous nucleotide mutations (or amino acid changes)?*

Response:

As the proposed analysis is based on amino acid data, phylogenetic analysis was also done using amino acid changes for consistency. The related text in the Methods section has been updated in the revised manuscript to make this point clear.

5. *Page 13, second paragraph: VDPVs are not exclusively obtained from evolution of the vaccine strain in immunised patients. Circulating VDPVs transmit from person to person in populations of low immunity so they could be very useful for assessing virus evolution under natural constraints. Indeed, it would be very interesting to compare evolution of wild-type versus circulating VDPVs.*

Response:

We agree that a comparison of the evolution of wild-type PV and circulating VDPVs would be interesting. However, as the focus of this work is on evolution of wild-type PV, we think that this comparison is best explored in an independent future study.

Reviewer #3 (Remarks to the Author):

The authors propose a method to infer an epistatic prevalence / fitness landscape for polioviruses protein vp1. Using this landscape, they map each observed vp1 sequence to a reference sequence defined as the local maximum in prevalence reached by steepest ascent. This mapping actually leads to a clustering of the originally more than 2000 sequences into 25 local landscape peaks, which in the paper are shown to contain highly interesting information in terms of geographical and temporal localisation of corresponding polio outbreaks. The most interesting point is probably the identification of a peak, which is likely related to the natural evolution of polioviruses over about 5 decades, while other peaks are related to the constrained evolution in a vaccinated population. Complementary analysis, e.g. comparison to in-vitro fitness measurements, further underline the significance of the found subdivision.

The article is very interesting and exceptionally clear in the presentation. Results are highly interesting, and the authors have investigated very carefully that the findings are not artefacts of their landscape inference procedure from finite data, or that the same results cannot be obtained easily by standard clustering procedures using phylogeny or machine learning methods like k-means or spectral clustering. Equally, non-epistatic landscape models as broadly used in computational biology are not able to capture the clustered structure of the data, so an epistatic modeling appears to be essential for finding the presented structuring of the data.

I have a number of pretty minor remarks, which should be taken into account to further strengthen the paper:

- 1. The paper finds 25 prevalence peaks, and I am slightly concerned about the robustness of this finding. The authors show that the inference based on finite data itself is probably not responsible for the peaks, by using an MSA with scrambled columns, which has an identical sequence profile but leads to a single peak. However, inference from finite data has typically two opposing tendencies. Finite sampling and overfitting tend to introduce roughness and thus more local peaks into the inferred landscape, and the authors provide a good even if not rigorous argument that this is not the case. On the other hand, inference from finite data typically requires the use of regularisation, which tends to make the landscape flatter and to merge local peaks. So how can we exclude that, e.g., peaks 2 and 4 are not a finite-data induced split of a single peak, and the important peak 1 is not a merger of slightly separated peaks due to regularisation? From the point of view of inference, it may be hard to exclude these scenarios, even if the subsequent analysis underlines the significance of the clustering procedure.*

As side remark, a network of inter-peak relations might be established in a non-stochastic way by looking to the minimal number of unfavourable mutations needed to overcome the prevalence valley between two peaks, or to study the stability of the peaks not only to single mutations but also to double-, triple mutations etc.

Response:

We fully agree that, like any model-based approach utilizing the available data, one cannot exclude the possibility of results being corrupted by finite sampling and overfitting issues. We were aware of these issues and we tried to tackle them as follows.

First, for demonstrating that the features of the inferred landscape are robust to errors in the empirical correlation from the MSA due to finite sampling, as correctly mentioned by the reviewer, we constructed a null case by shuffling the amino acids in each column of the MSA such that the number of amino acids observed at each residue remains the same, but the amino acids observed at different residues become uncorrelated. We fitted the Potts model to infer the landscape for this randomized case using the same procedure as discussed in Methods. In this case, only one local peak was observed in the inferred landscape, indicating that the multiple local peaks that we observe in the vp1 landscape are not an artifact of finite sampling. Second, as far as overfitting is concerned, our model inference procedure tries to avoid it as much as possible by selecting the model for which the error between the model correlations and the empirical correlations from the MSA is commensurate on average with the expected fluctuation in the correlations due to finite sampling (see (Barton, Kardar and Chakraborty, 2015) for further details). Third, we agree with the

reviewer that inferring a landscape from finite sample data involves the use of regularization, which can indeed affect the local peaks. In ACE, we used a L_2 -norm regularization parameter which was set to the recommended value of $\frac{1}{M}$, where M is the number of available PV sequences (Barton, De Leonardi, *et al.*, 2016). To test the robustness of the observed peaks to the value used for the regularization parameter, we re-inferred landscapes using a $\pm 50\%$ change in the recommended value of the parameter and found that the peaks remained almost the same (Fig. R12). We have included this analysis in the revised supplement (see Supplementary Text S3). Nevertheless, despite these statistical robustness checks, similar to other related works, we believe that the ultimate validation of the inferred landscape is the comparison with the available biological data. Our study of antigenic composition, temporal and geographical distribution of the different local peaks in the inferred landscape, and comparison with the experimental fitness values lends strong validation of our model and the associated findings.

Figure R12. Comparison of peaks obtained using different values of regularization parameter in landscape inference. Results are shown for the landscape inferred using (a) a 50% larger regularization parameter and (b) a 50% smaller regularization parameter than that one used in our landscape, respectively. The (i,j) th element of the matrix shows the percentage of sequences corresponding to the original peak present in the j th peak obtained using a different regularization parameter.

We agree that as opposed to the zero-temperature Monte Carlo approach that we used to study the inter-peak pathways, a non-stochastic approach as suggested by the reviewer can also be employed. However, this approach would be computationally intractable as it would require searching over the extraordinarily large number of mutational pathways that are possible from a given strain to reach a peak that is different from the corresponding peak of the strain. Thus, we believe that albeit stochastic, the zero-temperature Monte Carlo approach is a reasonable and efficient way to infer the possible inter-peak pathways.

As for defining peaks in a different way, indeed one can define these by following a double- (or higher-) mutation-based steepest ascent walk instead of the single-mutation-based steepest ascent walk that we used in our work. However, given the mutation rate of PV, the probability of observing two or more mutations in vp1 in successive generations during evolution is generally quite low. Thus, we believe that our current definition of peak sequences, i.e., using the single-mutation-based steepest ascent walk is reasonable. Nevertheless, to test the robustness of the proposed peaks to the type of steepest ascent walk used to define them, we re-defined peaks in our inferred vp1 landscape by following a double-mutation-based steepest ascent walk. In this case, the sequence in each next step was decided based on the mutation(s) that resulted in the highest increase in fitness over not only all single but also all double mutations possible in the current sequence. Using this procedure, instead of 25, we obtained 16 peaks (Fig. R13). Although some merging of peaks associated with largely immunized populations is observed in this case, the main insights of our

work related to peak 1, representing natural PV evolution in largely unimmunized populations, remained the same.

As a side remark, we would like to mention an interesting merging that was observed by defining peaks using the double-mutation-based steepest ascent walk. Specifically, all sequences corresponding to peaks 2, 4, and 6 were merged into peak 2 (Fig. R13). While the sequences in both peaks 2 and 4 represent primarily Egypt outbreaks and thus this merging was somewhat expected, the merging of peak 6 (comprising sequences largely from multiple South American countries and Israel) in peak 2 was less clear. Looking at the associated literature reports, it seems that the outbreaks represented by peak 6, that occurred in the largely immunized population of South America in 1980 and Israel in 1988, may have been caused by the importation of wild-type virus from Egypt (Shulman *et al.*, 2000; Jorba *et al.*, 2008). Nonetheless, as mentioned above, this merging of peaks associated with immunized populations does not change the main message of our work related to disentangling the natural PV evolution in largely unimmunized populations from that under the strong effects of the (IPV/OPV) vaccine.

Figure R13. Comparison of peaks obtained using the single- and double-mutation-based steepest ascent walks. The $(i,j)^{th}$ element of the matrix shows the percentage of sequences corresponding to the j^{th} single-mutation-based peak present in the i^{th} double-mutation-based peak.

- Concerning the enrichment of mutations in the antigenic sites, a better explication of the null model / p-value might be helpful. Is the calculation equivalent with doing a large number of random selections of residues and showing that the antigenic sites have more mutations than a random selection? Or does it include some implicit randomisation of the mutational patterns?

Response:

In Fig. 2c, the statistical significance of the observed fraction of mutations on antigenic sites in each peak sequence was quantified using a p-value, which is the probability of observing a result as extreme as or more extreme than the one being studied, assuming a null hypothesis were true. For example, assume that there are j antigenic sites in vp1 and that a peak sequence, comprising n

mutations, includes mutations on i of the j antigenic sites. Here, the null hypothesis would be that the observed mutations on the n sites in this peak sequence arose from a random selection from the N sites of the protein. Assuming that the null hypothesis is true, the p-value is the probability that a peak sequence would include mutations on at least i of the j antigenic sites and is calculated as follows:

$$p = \sum_{q=i}^{\min(j,n)} \frac{\binom{j}{q} \binom{N-j}{n-q}}{\binom{N}{n}} .$$

A low p-value ($p < 0.05$) would indicate that the null hypothesis is rejected and that it is unlikely that such a peak could arise from random chance.

We have now elaborated the description of the p-value calculation in the revised manuscript.

3. Are the statistical energies of the peak sequences very different from the natural sequences, or in between each other? Are they informative about the significance of the cluster?

Response:

The predicted energies of the peak sequences are much lower than those of the natural sequences ($P = 5.8 \times 10^{-3}$; two-tailed Mann-Whitney test) (Fig. R14). This is expected as the peak sequences are associated with the local minima (or local maxima with respect to prevalence/fitness) in the landscape. However, it is not clear if the energy of a peak sequence provides any information about the significance of the associated sequences over other peaks. This is due to the fact that all peaks, except peak 1, represent specific temporally and geographically localized outbreaks (as discussed in the Results section of the manuscript).

Figure R14. Comparison of the distribution of energies of all available vp1 sequences and peak sequences.

4. The ρ_s with in vitro fitness measurements for the full MSA was indicated as -0.64, but it is not written if these concerns all measurements or the ones without outliers.

Response:

The reported value of the Spearman correlation between experimental fitness measurements and in-silico predicted energy values obtained from the landscape inferred using all sequences except peak 1 (Supplementary Fig. S4) included the outliers. However, this point is no longer relevant as the data used to generate Supplementary Fig. S4 has been changed in the revised version for addressing the concerns raised by reviewer 2.

5. Concerning the terminology, "low-dimensional decomposition" sounds more like PCA than discrete finite clustering.

We agree that the terminology "low-dimensional decomposition" is more related to PCA. However, as PCA is also used for clustering related sequences (e.g., see Figure S8), we think it is suitable for describing our peak analysis as a low-dimensional representation of sequence space.

6. Concerning Fig. 1, the fitting and prediction qualities look impressive. However, to fully appreciate the power of the used epistatic model, it would be better to use connected correlations than double- or triple-mutant frequencies. To explain my point, if the single-site frequencies are biased, even in a model with independent sites there are non-trivial two- and three-site frequencies, which however can be reproduced by a profile model. Nonzero connected correlations cannot.

Response:

We agree and have replaced the two- and three-point correlations ($p_{ij}(x, y)$ and $p_{ijk}(x, y, z)$, respectively) with the corresponding connected correlations ($p_{ij}(x, y) - p_i(x)p_j(y)$ and $p_{ijk}(x, y, z) - p_i(x)p_{jk}(y, z)$, respectively) in Fig. 1. The inferred model accurately predicts these connected correlations of the MSA as well (Fig. 1b-1c).

Figure 1. The inferred Potts model accurately captures the one-point and two-point mutational correlations, as well as the higher order statistics of the observed sequences.

(a-b) Comparison of the data statistics used to train the model: (a) The one-point mutational correlations, $p_i(x)$, and (b) the two-point mutational connected correlations, $p_{ij}(x, y) - p_i(x)p_j(y)$, of the observed sequences and the those obtained from the inferred Potts model. (c-f) Comparison of the data statistics predicted by the inferred Potts model: (c) The three-point mutational connected correlations (which represents how mutations at two residues influence the probability of mutation at a third residue, i.e., $p_{ijk}(x, y, z) - p_i(x)p_{jk}(y, z)$), (d) the distribution of the number of mutations, (e) the distribution of the first most-prevalent mutant amino acid, and (f) the distribution of the second most-prevalent mutant amino acid in the observed sequences and the inferred Potts model.

7. To conclude, I think that the presented work is very interesting and well written, and only minor revisions would be needed.

Response:

We appreciate the positive response and the constructive review, which has led to improvement of our paper.

References

- Acevedo, A., Brodsky, L. and Andino, R. (2013) 'Mutational and fitness landscapes of an RNA virus revealed through population sequencing.', *Nature*. Nature Publishing Group, 505(7485), pp. 686–90. doi: 10.1038/nature12861.
- Barton, J. P., De Leonardis, E., *et al.* (2016) 'ACE: adaptive cluster expansion for maximum entropy graphical model inference - biorxiv', *Bioinformatics*, 32(20), pp. 3089–3097. doi: 10.1093/bioinformatics/btw328.
- Barton, J. P., Goonetilleke, N., *et al.* (2016) 'Relative rate and location of intra-host HIV evolution to evade cellular immunity are predictable', *Nature Communications*. Nature Publishing Group, 7, p. 11660. doi: 10.1038/ncomms11660.
- Barton, J. P., Kardar, M. and Chakraborty, A. K. (2015) 'Scaling laws describe memories of host–pathogen riposte in the HIV population', *Proceedings of the National Academy of Sciences*, 112(7), pp. 1965–1970. doi: 10.1073/pnas.1415386112.
- El Bassioni, L. *et al.* (2003) 'Prolonged detection of indigenous wild polioviruses in sewage from communities in Egypt', *American Journal of Epidemiology*, 158(8), pp. 807–815. doi: 10.1093/aje/kwg202.
- Bouchard, M. J., Lam, D. and Racaniello, V. R. (1995) 'Determinants of attenuation and temperature sensitivity in the type 1 poliovirus Sabin vaccine', *Journal of virology*, 69(8), pp. 4972–4978.
- Butler, T. C. *et al.* (2016) 'Identification of drug resistance mutations in HIV from constraints on natural evolution', *Physical Review E*, 93(2), pp. 1–8. doi: 10.1103/PhysRevE.93.022412.
- Chakraborty, A. K. and Barton, J. P. (2017) 'Rational design of vaccine targets and strategies for HIV: a crossroad of statistical physics, biology, and medicine', *Reports on Progress in Physics*. IOP Publishing, 80(3), p. 032601. doi: 10.1088/1361-6633/aa574a.
- Cocco, S., Monasson, R. and Weigt, M. (2013) 'From principal component to direct coupling analysis of coevolution in proteins: Low-eigenvalue modes are needed for structure prediction', *PLoS Computational Biology*, 9(8), p. e1003176. doi: 10.1371/journal.pcbi.1003176.
- Colston, E. M. and Racaniello, V. R. (1995) 'Poliovirus variants selected on mutant receptor-expressing cells identify capsid residues that expand receptor recognition.', *Journal of virology*, 69(8), pp. 4823–4829.
- Colston, E. and Racaniello, V. R. (1994) 'Soluble receptor-resistant poliovirus mutants identify surface and internal capsid residues that control interaction with the cell receptor.', *The EMBO journal*, 13(24), pp. 5855–5862.
- Couderc, T. *et al.* (1993) 'Molecular characterization of mouse-virulent poliovirus type 1 Mahoney mutants: involvement of residues of polypeptides VP1 and VP2 located on the inner surface of the capsid protein shell.', *Journal of virology*, 67(7), pp. 3808–3817.
- Couderc, T. *et al.* (1994) 'Substitutions in the capsids of poliovirus mutants selected in human neuroblastoma cells confer on the Mahoney type 1 strain a phenotype neurovirulent in mice.', *Journal of virology*, 68(12), pp. 8386–8391.
- Dahirel, V. *et al.* (2011) 'Coordinate linkage of HIV evolution reveals regions of immunological vulnerability', *Proceedings of the National Academy of Sciences*, 108(28), pp. 11530–11535. doi: 10.1073/pnas.1105315108.
- Drexler, J. F. *et al.* (2014) 'Robustness against serum neutralization of a poliovirus type 1 from a lethal epidemic of poliomyelitis in the Republic of Congo in 2010.', *Proceedings of the National Academy of Sciences of the United States of America*, 111(35), pp. 12889–94. doi: 10.1073/pnas.1323502111.
- Ferguson, A. L. *et al.* (2013) 'Translating HIV sequences into quantitative fitness landscapes predicts viral vulnerabilities for rational immunogen design', *Immunity*, 38(3), pp. 606–617. doi: 10.1016/j.immuni.2012.11.022.
- Figliuzzi, M. *et al.* (2016) 'Coevolutionary landscape inference and the context-dependence of mutations in beta-lactamase TEM-1', *Molecular Biology and Evolution*, 33(1), pp. 268–280. doi:

10.1093/molbev/msv211.

Flynn, W. F. *et al.* (2017) 'Inference of epistatic effects leading to entrenchment and drug resistance in HIV-1 protease', *Molecular Biology and Evolution*, 34(6), pp. 1291–1306. doi:

10.1093/molbev/msx095.

Green, M. S. *et al.* (1993) 'Age differences in immunity against wild and vaccine strains of poliovirus prior to the 1988 outbreak in Israel and response to booster immunization', *Vaccine*, 11(1), pp. 75–81. doi: 10.1016/0264-410X(93)90342-U.

Hart, G. R. and Ferguson, A. L. (2015) 'Empirical fitness models for hepatitis C virus immunogen design', *Physical Biology*. IOP Publishing, 12(6), p. 066006. doi: 10.1088/1478-3975/12/6/066006.

Hinkley, T. *et al.* (2011) 'A systems analysis of mutational effects in HIV-1 protease and reverse transcriptase.', *Nature genetics*, 43(5), pp. 487–489. doi: 10.1038/ng.795.

Hogle, J. M. and Filman, D. J. (1989) 'The antigenic structure of poliovirus', *Philosophical Transactions of the Royal Society of London. Series B, Biological Sciences*, 323(1217), pp. 467–478.

Hopf, T. A. *et al.* (2017) 'Mutation effects predicted from sequence co-variation', *Nature biotechnology*. Nature Publishing Group, 35(2), pp. 128–135. doi: 10.1038/nbt.3769.

Jorba, J. *et al.* (2008) 'Calibration of multiple poliovirus molecular clocks covering an extended evolutionary range.', *Journal of virology*, 82(9), pp. 4429–4440. doi: 10.1128/JVI.02354-07.

Kojouharova, M. *et al.* (2003) 'Importation and circulation of poliovirus in Bulgaria in 2001', *Bulletin of the World Health Organization*, 81(7), pp. 476–481.

Kryazhimskiy, S. and Plotkin, J. B. (2008) 'The population genetics of dN/dS.', *PLoS genetics*, 4(12), p. e1000304. doi: 10.1371/journal.pgen.1000304.

Kumar, S., Stecher, G. and Tamura, K. (2016) 'MEGA7: Molecular Evolutionary Genetics Analysis Version 7.0 for Bigger Datasets', *Molecular biology and evolution*, 33(7), pp. 1870–1874. doi: 10.1093/molbev/msw054.

Li, W.-H., Wu, C.-I. and Luo, C.-C. (1985) 'A new method for estimating synonymous and nonsynonymous rates of nucleotide substitution considering the relative likelihood of nucleotide and codon changes.', *Molecular Biology and Evolution*, 2(2), pp. 150–174. doi:

10.1093/oxfordjournals.molbev.a040343.

Liao, S. and Racaniello, V. (1997) 'Allele-specific adaptation of poliovirus VP1 B-C loop variants to mutant cell receptors.', *Journal of virology*, 71(12), pp. 9770–9777.

Liu, H., Zheng, D. and Zhang, L. (2000) 'Molecular evolution of a type 1 wild-vaccine poliovirus recombinant during widespread circulation in China', *Journal of virology*, 74(23), pp. 11153–11161. doi: 10.1128/JVI.74.23.11153-11161.2000.

Louie, R. H. Y. *et al.* (2018) 'Fitness landscape of the human immunodeficiency virus envelope protein that is targeted by antibodies', *Proceedings of the National Academy of Sciences*, 115(4), pp. E564–E573. doi: 10.1073/pnas.1717765115.

Mann, J. K. *et al.* (2014) 'The fitness landscape of HIV-1 Gag: Advanced modeling approaches and validation of model predictions by in vitro testing', *PLoS Computational Biology*. Edited by R. R. Regoes, 10(8), p. e1003776. doi: 10.1371/journal.pcbi.1003776.

Mihalek, I., Reš, I. and Lichtarge, O. (2004) 'A Family of Evolution-Entropy Hybrid Methods for Ranking Protein Residues by Importance', *Journal of Molecular Biology*, 336(5), pp. 1265–1282. doi: 10.1016/j.jmb.2003.12.078.

Morcos, F. *et al.* (2011) 'Direct-coupling analysis of residue coevolution captures native contacts across many protein families.', *Proceedings of the National Academy of Sciences of the United States of America*, 108(49), pp. E1293–E1301. doi: 10.1073/pnas.1111471108.

Odoom, J. K. *et al.* (2012a) 'Interruption of poliovirus transmission in Ghana: Molecular epidemiology of wild-type 1 poliovirus isolated from 1995 to 2008', *Journal of Infectious Diseases*, 206(7), pp. 1111–1120. doi: 10.1093/infdis/jis474.

Odoom, J. K. *et al.* (2012b) 'Interruption of poliovirus transmission in Ghana: Molecular epidemiology of wild-type 1 poliovirus isolated from 1995 to 2008', *Journal of Infectious Diseases*, 206(7), pp. 1111–1120. doi: 10.1093/infdis/jis474.

Qin, C. and Colwell, L. J. (2018) 'Power law tails in phylogenetic systems', *Proceedings of the National Academy of Sciences*, 115(4), pp. 690–695. doi: 10.1073/pnas.1711913115.

Quadeer, A. A., Morales-Jimenez, D. and McKay, M. R. (2018) 'Co-evolution networks of HIV/HCV are modular with direct association to structure and function', *PLoS Computational Biology*. Edited by R. D. Kouyos, 14(9), p. e1006409. doi: 10.1371/journal.pcbi.1006409.

Rausell, A. *et al.* (2010) 'Protein interactions and ligand binding: from protein subfamilies to functional specificity.', *Proceedings of the National Academy of Sciences of the United States of America*, 107(5), pp. 1995–2000. doi: 10.1073/pnas.0908044107.

Shaukat, S. *et al.* (2014) 'Molecular characterization and phylogenetic relationship of wild type 1

poliovirus strains circulating across Pakistan and Afghanistan bordering areas during 2010–2012', *PLoS ONE*, 9(9), p. e107697. doi: 10.1371/journal.pone.0107697.

Shulman, L. M. *et al.* (2000) 'Resolution of the pathways of poliovirus type 1 transmission during an outbreak', *Journal of Clinical Microbiology*, 38(3), pp. 945–952.

Shulman, L. M. *et al.* (2015) 'Genetic analysis and characterization of wild poliovirus type 1 during sustained transmission in a population with 95% vaccine coverage, Israel 2013', *Clinical Infectious Diseases*, 60(7), pp. 1057–1064. doi: 10.1093/cid/ciu1136.

Sutter, R. W. *et al.* (2018) *Poliovirus Vaccine–Live*. Seventh Ed, *Plotkin's Vaccines*. Seventh Ed. Elsevier Inc. doi: 10.1016/B978-0-323-35761-6.00048-1.

World-Health-Organization (2016) *Poliomyelitis, Fact sheet*. Available at: <http://www.who.int/mediacentre/factsheets/fs114/en/> (Accessed: 20 July 2017).

Yakovenko, M. L. *et al.* (2014) 'The 2010 outbreak of poliomyelitis in Tajikistan: Epidemiology and lessons learnt.', *Eurosurveillance*, 19(7), p. 20706. doi: 10.2807/1560-7917.ES2014.19.7.20706.

Yusuf, N. *et al.* (2014) 'Outbreak of type 1 wild poliovirus infection in adults, Namibia, 2006', *Journal of Infectious Diseases*, 210(suppl 1), pp. S353–S360. doi: 10.1093/infdis/jiu069.

Reviewers' comments:

Reviewer #1 (Remarks to the Author):

The authors have made a length documentation of the work they have done to address the comments made by the reviewers. I was happy with their response on my biggest concern, which is that of biases due to sampling of sequences, i.e. phylogenetic relationships. I still do see two main concerns. First, since this journal is a general journal, I believe it will be very hard for readers to follow. The response letter was long and I am not sure the way this is written in the paper is easy enough to follow. Second, regarding mutational coupling, which I agree is an advantage of the approach. Which couplings are responsible for the signal they are getting? Can they verify some biological meaning for these? While I realize other methods do not account for coupling, I believe the authors can find other methods to compare their approach with and point out the biological validity. Otherwise this is just a hidden feature that the reader is asked to trust blindly.

Reviewer #2 (Remarks to the Author):

The paper by Ahmed A. Quadeer et al. present an elegant model to show that poliovirus evolution is subject to tight intrinsic fitness constraints that limit its capacity to evade vaccine-induced immune responses.

The authors have satisfactorily address most of my queries although some explanations are still speculative and based on a small number of sequences. I am still unsure about how some of the virus data from the literature (Table S2) were used to generate Figure 5. What were the criteria to use information from some mutants described in these papers and not from others? The correct figure for the titer of mutants described in the Colston and Racaniello 1994 paper is 4.4×10^6 and not 4.4×10^4 ?

Reviewer #3 (Remarks to the Author):

Already my first report was very positive because of the high interest and the clear presentation of the results in the manuscript. I had raised only minor issues. Despite their minority, the authors have given very detailed responses, and modified the paper accordingly.

I am totally satisfied with their revision, and strongly recommend publication of the revised manuscript.

RESPONSE TO REVIEWERS' COMMENTS

Deconvolving mutational patterns of poliovirus outbreaks reveals its intrinsic fitness landscape (NCOMMS-18-08369A-Z)

We thank the reviewers for the time they devoted to provide detailed and thoughtful reviews, which have helped to improve our paper. Below, we address each of the reviewers' comments and note the changes made to the manuscript and the supplement in response (in blue).

Reviewer #1 (Remarks to the Author):

The authors have made a length documentation of the work they have done to address the comments made by the reviewers. I was happy with their response on my biggest concern, which is that of biases due to sampling of sequences, i.e. phylogenetic relationships. I still do see two main concerns.

Response:

Thank you for the time and effort in reviewing our paper. We are glad that the reviewer appreciated our response to the major concerns raised in the previous review, which we believe has led to substantial improvement to our work. Our point-by-point responses to the two further concerns raised by the reviewer are detailed below.

1. *First, since this journal is a general journal, I believe it will be very hard for readers to follow. The response letter was long and I am not sure the way this is written in the paper is easy enough to follow.*

Response:

We understand the reviewer's concern regarding readability of the manuscript. To address it, we have carefully gone over the manuscript and have made multiple changes throughout (shown in blue color in the revised version). Particularly, we have now substantially revised the abstract and Fig. 1, removed/simplified the technical or field-specific terminology (which assumed reader to have specific domain knowledge) in the main text, and have made sure the consistent use of terms throughout the manuscript. We believe these changes have helped in making the manuscript easy to follow for the general audience of this journal.

2. *Second, regarding mutational coupling, which I agree is an advantage of the approach. Which couplings are responsible for the signal they are getting? Can they verify some biological meaning for these? While I realize other methods do not account for coupling, I believe the authors can find other methods to compare their approach with and point out the biological validity. Otherwise this is just a hidden feature that the reader is asked to trust blindly.*

Response:

We appreciate the concern of the reviewer and understand that the interpretation of the model couplings, and potential biological meaning, could be further clarified. Here we provide such interpretation, supported by additional tests, and describe the changes we have made to the manuscript accordingly.

There are two sets of couplings that should be considered: those in the original model based on all sequences, and those in the model based on peak-1 sequences only, which we propose as the PV fitness landscape model. We address each of these in turn.

Regarding the first (all-sequence) model, we recall that incorporating couplings using pairwise residue variation is *necessary* to observe the peak structure, since only a single peak is observed if a model is inferred using single-residue variation only (details in Supplementary Text S1 and S2). Moreover, as we emphasize in the Results section, the analysis of the peak structure was crucial to disentangle the natural PV evolution from the vaccine-driven (outbreaks-associated) evolution, which then allowed us to suppress the bias in the available PV sequence data caused by outbreaks and obtain a meaningful PV fitness landscape.

Examining the strongest couplings in the model, corresponding to the 1% of those with the highest magnitudes, reveals that each observed peak is enriched with such couplings. Many of these couplings also involve the known biologically-important vp1 *antigenic sites* (Fig. R1-1). This is consistent with the suggested role of antigenic mutations in escaping the immune pressure and triggering outbreaks (Figs. 2b, 2c). This has been discussed in ref¹ in the context of the 2011 Congo outbreak (represented by peak 3 in our model), when antigenic mutations in the circulating strain resulted in a large number of fatalities.

Figure R1-1. Couplings among mutations in the peak sequences are ranked among the top 1 percentile of the all-sequence based model couplings. For each peak, only the couplings among the mutations in the peak sequence were considered. The peak numbers are shown above each subfigure. The couplings that are ranked in the top 1 percentile of the inferred model couplings are shown as connected. Connections involving at least one known vp1 antigenic site are shown in green color while all remaining ones are shown in black color. Results are shown for peaks 2 to 10.

More generally, the peak structure induced by the couplings appears to be a manifestation of biased correlations resulting from localized variation observed in specific outbreaks that is tied to antigenic variation. That is, sequences in outbreaks tend to be closely related, sharing highly specific sets of mutations. We conducted additional tests to demonstrate this, by showing that mutations in each “peak sequence” in the model (i.e., the most fit, or lowest energy, sequence in a peak), referred to as “peak mutations” henceforth, were shared by a large majority of sequences falling in the same peak, inducing strong localized correlations (Fig. R1-2). In contrast, other mutations in the sequences falling in the same peak, referred to as “non-peak mutations” here onwards, occurred at a much lower frequency and consequently induced much weaker correlations (Fig. R1-2).

Figure R1-2. Correlations among peak mutations in each peak are stronger than the corresponding non-peak mutations. The (a) single and (b) double mutational probabilities were computed for each peak by considering all sequences falling in that peak. Results are shown for peaks 2 to 10 only. Error bars represent one standard error.

From an evolutionary perspective, the strong couplings observed among peak mutations may be seen to arise due to “genetic linkage”. (That is, due to the over-representation of background variation coupled to antigenic variation offering a selective advantage.) Conceptually, this phenomenon is similar to that observed in the evolution of influenza virus^{2,3}, in which multiple neutral or deleterious mutations linked to antigenic mutations often collectively rise to fixation. To further quantify the effect of genetic linkage in shaping the outbreak-associated peaks, we computed the linkage disequilibrium⁴ in the sequences associated with each peak using the standard Lewontin D' metric⁵ (Fig. R1-3). This analysis suggested that all of the top 10 peaks, except peak 1, were seemingly strongly affected by genetic linkage (mean $D' \sim 1$).

Figure R1-3. Comparison of genetic linkage associated with the sequences corresponding to each peak. Genetic linkage was quantified using the standard Lewontin D' measure⁵. The height of each bar represents the mean D' value across all the pairs of mutations in the sequences corresponding to that peak. Error bars represent one standard error.

The coupling structure for the model built from sequences associated with largely unimmunized population (represented by peak 1) is completely different, and the landscape has only a single peak, as we have described. The majority of the residue pairs linked by strong couplings in this

model were distinct from those in the all-sequence model (Fig. R1-4), and there is no longer a strong enrichment of antigenic mutations in this set. The linkage disequilibrium in the sequences associated with peak 1 is also substantially less (Fig. R1-3). This supports the notion that the second model is subject to significantly lower sequence bias than the all-sequence model, and this leads to a model which is a meaningful proxy for the vp1 fitness landscape, as we demonstrate through comparison with independent experimental fitness data.

Figure R1-4. Majority of couplings among peak mutations are not ranked among the top 1 percentile of the model couplings inferred using peak 1 sequences only. Note that the peaks are defined based on the all-sequence model. The peak numbers are shown above each subfigure. For each peak, only the couplings among the respective peak mutations are studied. The couplings that are ranked in the top 1 percentile of the inferred model couplings are shown as connected. The couplings between pairs involving any antigenic site are shown in green color while all remaining ones are shown in black color. Results are shown for peaks 2 to 10.

For peak 1, the couplings are statistically smaller in magnitude ($P = 10^{-32}$; Mann-Whitney test) as compared to the model inferred in the all-sequence model (Fig. R1-5). Note that this result is consistent with the known reduced role of epistasis in PV evolution^{6,7}. However, investigating the top 1 percentile model couplings shows that 48% of these involve N terminus vp1 residues which form an interface with the capsid vp4 protein (Fig. R1-6) known to be critical for viral stability⁸. This suggests that many of the strongest couplings encode information about constraints on the vp1-vp4 protein structure. That is, coupled mutations affect viral fitness together due to structural reasons. This result sheds direct light on the biological meaning of the intrinsic fitness landscape that we have inferred after carefully removing sampling biases due to localized outbreaks, vaccination, etc.

Fig. R1-5. Comparison of the couplings inferred in the all-sequence and the peak 1 based model.

Fig. R1-6. A large proportion of the top 1 percentile couplings in the peak 1 model are involved in forming a critical vp1-vp4 interface. The vp1 and vp4 pentamers are shown as blue and green spheres respectively, while the vp1 residues (in the top 1 percentile model couplings) involved in the vp1-vp4 interface are shown as red spheres. The crystal structure of the poliovirus capsid was downloaded from the PDB database, <https://www.rcsb.org/> (ID: 2PLV).

To summarize, the inferred couplings in the original model are vital for suppressing the strong sampling bias in the available PV sequence data caused by outbreaks, which then enabled us to obtain a meaningful PV fitness landscape. The strong couplings in this model are enriched in antigenic mutations which seemingly facilitate outbreaks. In contrast, the couplings in the model built from only peak 1 sequences (representing the PV fitness landscape) seem to be biologically informative from a structural/virus fitness, and is consistent with the reported limited role of interactions between residues in PV evolution^{6,7}.

We have revised the text related to interpretation of model couplings in the Discussion section (see text in blue below) and included the above results in a new supplementary section (S8).

Reviewer #2 (Remarks to the Author):

The paper by Ahmed A. Quadeer et al. present an elegant model to show that poliovirus evolution is subject to tight intrinsic fitness constraints that limit its capacity to evade vaccine-induced immune responses.

The authors have satisfactorily address most of my queries although some explanations are still speculative and based on a small number of sequences. I am still unsure about how some of the virus data from the literature (Table S2) were used to generate Figure 5. What were the criteria to use information from some mutants described in these papers and not from others? The correct figure for the titer of mutants described in the Colston and Racaniello 1994 paper is 4.4×10^6 and not 4.4×10^4 ?

Response

We appreciate the potential confusion related to the criteria we used to select the specific experimental fitness measurements (listed in Table S2) from each report⁹⁻¹³. We have now tried to elaborate this in detail for each report as discussed below.

In ref¹⁰, the authors compared in Table I the growth (in PFU/ml) of Mahoney and soluble receptor-resistant (srr) mutants in the presence and absence of soluble poliovirus receptor (S100Pvr). As our model is trained based on *vp1 sequences* evolving in the population with the *wild-type Pvr*, we only selected the virus titers reported in the absence of S100Pvr. The srr mutants in the vp1 protein are listed in Table II. Note that as all mutants were found to have similar titers, only the average titer value

for srr mutants was reported in Table I. Thus, for fair comparison, we also averaged over the energies (predicted from our model) corresponding to all mutant strains. In Fig. 5a, only two data points are associated with this report: one for the Mahoney strain and the other for the averaged srr mutants' strains. The binding affinities and alteration ratios of srr mutants, reported in Tables III and IV respectively, were excluded from our analysis as we considered viral titers (Table I) to be a better representative of the viral replicative fitness. We thank the reviewer for pointing out the typographical error associated with this report in Table S2. The correct titer value for mutants is 4.4×10^6 PFU/ml.

In ref¹¹, the authors compared in Table 1 the growth (in PFU/ml) of wild-type and mutant PV (serotypes 1, 2, and 3) on cells expressing wild-type and mutant (d, g, and i) Pvr. As our model is trained based on *PV serotype 1 vp1 sequences* evolving in the population with the *wild-type Pvr*, we selected only the virus titers reported for Mahoney (serotype 1 wild-type PV) and the associated vp1 mutant strains on cells expressing the wild-type Pvr. Moreover, note that the measurement for the strain with P95S mutant was selected while that with P95T mutant was not because we only observed one mutation (P to S) at position 95 in the available vp1 sequence data. Also, the binding affinities reported in Table 3 were not used for the same reason as mentioned above for ref¹⁰.

In ref¹², the virus titers of wild-type and mutant poliovirus determined by plaque assays on HeLa cell monolayers at 37 degrees Celsius as well as a slightly higher temperature of 40 degrees Celsius were reported in Table 2. These titers were reported for the Mahoney strain, the Sabin strain, multiple recombinant strains of Mahoney and Sabin strains, and multiple mutant Mahoney strains with mutations in different viral proteins. We only considered the eight virus titers related to our work, i.e., those corresponding to the Mahoney strain and its mutants in the vp1 protein (listed in Table S2). We excluded the Sabin strain as the vaccine-related strains were not included in the data used for inferring our model (see Fig. S8 in the revised supplement). Also, following the suggestion of the reviewer in the previous review that virus titers at high temperature correlate better with data from human isolates, we included in our analysis the virus titers determined at higher temperature (40 degrees Celsius) in Table 2. Moreover, as mentioned in the previous response, we did not include the 50% paralytic/lethal dose (PLD50) values reported in Table 1 of ref¹² as these did not seem to correlate well (Fig. R2-1; reproduced below for convenience) with the reported titers for strains with the same mutations in Table 2 of ref¹².

In ref¹³, the authors reported in Table 2 the growth (in PFU/ml) of (i) $\Delta 9$ strain: a B-C loop (residues 98–102) truncated Mahoney strain, (ii) 414 strain: Mahoney strain with B-C loop replaced with the corresponding sequence in the serotype 2 Lansing strain, and (iii) multiple variants of $\Delta 9$ and 414 strains that are adapted to three Pvr mutant cell lines (d, g, and i). All the virus titers reported for the latter multiple variants (Tables 4, 6, 7, and 8), except the variant $\Delta 9$ -1/i, were not considered as these adapted viruses included mutations in proteins other than vp1 (shown in Table 3 of ref¹³). The variant $\Delta 9$ -1/i included three mutants in vp1 and the growth of strains having all possible (seven) combinations of these mutants were studied by site-directed mutagenesis in $\Delta 9$ strain on d, g, i, and 20B cells in Table 5 of ref¹³. As in the case of ref¹¹ (mentioned above), we only considered the virus titers reported in Table 5 for 20B cells as these express wild-type Pvr. The titer for $\Delta 9$ -1/i reported in Table 5 of ref¹³ was not included as it is essentially similar to the titer reported for the $\Delta 9$ strain with all the three mutants. Moreover, we had previously excluded the 414 strain as it involved a part of the serotype 2 Lansing strain in vp1. However, we investigated further and found that our model predictions are robust to the inclusion of the virus titer of the 414 strain and the associated strain with mutation (E168G) in vp1 (reported in Table 2 and 8, respectively) in the analysis (Fig. R2-2). Note that other virus titers reported in Table 8 could not be included as these involved mutations in proteins other than vp1.

In ref⁹, the neurovirulence, in terms of 50% paralytic dose (PD50), of multiple strains in transgenic mice with wild-type Pvr was reported in Table 2. Of these strains, we did not include the measurements reported for the Sabin strain for the same reason as above for ref¹². The measurements reported for the strains 02v9529 and PAK-5388 were also excluded as the authors did not provide the accession numbers associated with them. (02v9529 appeared to be a vaccine-derived strain¹⁴ and thus not related to our analysis). The remaining PD50 measurements reported in Table 2 of ref⁹ for the Mahoney strain and the three Israel 2013 outbreak-associated strains were included in our analysis. Data reported in Table 3 of ref⁹ was not included as it was related to neutralization potency of antisera from rats immunized with IPV.

As can be seen from the above discussion, the experimental fitness measurements listed in Table S2 were obtained after an extensive survey of reports present in the literature. While we agree that the

total number of measurements is limited, we believe that the small p-value ($\sim 10^{-8}$) associated with the result (Fig. 5a) indicates strong evidence against obtaining such a high correlation between model predictions and experimental fitness measurements (-0.83) purely by chance.

We have included the above important details related to selecting experimental fitness measurements from literature for model validation in a new supplementary section (S7).

Figure R2-1. Normalized virus titers vs normalized PLD50 values reported in ref¹² for the same viral strains.

Figure R2-2. Comparison of in silico predicted energy and experimental fitness measurements after including the virus titers related to the 414 strain reported in ref¹³. Both the fitness measurements and the predicted energies have been normalized using the standard procedure of subtracting the mean from each data set and dividing by its standard deviation.

Reviewer #3 (Remarks to the Author):

Already my first report was very positive because of the high interest and the clear presentation of the results in the manuscript. I had raised only minor issues. Despite their minority, the authors have given very detailed responses, and modified the paper accordingly.

I am totally satisfied with their revision, and strongly recommend publication of the revised manuscript.

Response

We appreciate the extremely positive response of the reviewer and thank him/her again for the time and effort spent in reviewing our work.

References

1. Drexler, J. F. *et al.* Robustness against serum neutralization of a poliovirus type 1 from a lethal epidemic of poliomyelitis in the Republic of Congo in 2010. *Proc. Natl. Acad. Sci. U. S. A.* **111**, 12889–94 (2014).
2. Neverov, A. D., Kryazhimskiy, S., Plotkin, J. B. & Bazykin, G. A. Coordinated evolution of influenza A surface proteins. *PLOS Genet.* **11**, e1005404 (2015).
3. Strelkova, N. & Lässig, M. Clonal interference in the evolution of influenza. *Genetics* **192**, 671–82 (2012).
4. Slatkin, M. Linkage disequilibrium--understanding the evolutionary past and mapping the medical future. *Nat. Rev. Genet.* **9**, 477–485 (2008).
5. Lewontin, R. C. The interaction of selection and linkage. I. General considerations; heterotic models. *Genetics* **49**, 49–67 (1964).
6. Burch, C. L., Turner, P. E. & Hanley, K. A. Patterns of epistasis in RNA viruses: A review of the evidence from vaccine design. *J. Evol. Biol.* **16**, 1223–1235 (2003).
7. de Visser, J. A. G. M. & Elena, S. F. The evolution of sex: empirical insights into the roles of epistasis and drift. *Nat. Rev. Genet.* **8**, 139–149 (2007).
8. Hogle, J. M. Poliovirus Cell Entry: Common Structural Themes in Viral Cell Entry Pathways. *Annu. Rev. Microbiol.* **56**, 677–702 (2002).
9. Shulman, L. M. *et al.* Genetic analysis and characterization of wild poliovirus type 1 during sustained transmission in a population with 95% vaccine coverage, Israel 2013. *Clin. Infect. Dis.* **60**, 1057–1064 (2015).
10. Colston, E. & Racaniello, V. R. Soluble receptor-resistant poliovirus mutants identify surface and internal capsid residues that control interaction with the cell receptor. *EMBO J.* **13**, 5855–5862 (1994).
11. Colston, E. M. & Racaniello, V. R. Poliovirus variants selected on mutant receptor-expressing cells identify capsid residues that expand receptor recognition. *J. Virol.* **69**, 4823–4829 (1995).
12. Bouchard, M. J., Lam, D. & Racaniello, V. R. Determinants of attenuation and temperature sensitivity in the type 1 poliovirus Sabin vaccine. *J. Virol.* **69**, 4972–4978 (1995).
13. Liao, S. & Racaniello, V. Allele-specific adaptation of poliovirus VP1 B-C loop variants to mutant cell receptors. *J. Virol.* **71**, 9770–9777 (1997).
14. Martin, J. *et al.* Long-term excretion of vaccine-derived poliovirus by a healthy child. *J. Virol.* **78**, 13839–13847 (2004).

REVIEWERS' COMMENTS:

Reviewer #1 (Remarks to the Author):

I have no further comments!

Reviewer #2 (Remarks to the Author):

I am satisfied with the revision by the authors